# Host heparan sulfate promotes ACE2 super-cluster assembly and enhances SARS-CoV-2-associated syncytium formation

Qi Zhang [1,2], Weichun Tang[3], Eduardo Stancanelli[4], Eunkyung Jung[5], Zulfeqhar Syed [6], Vijayakanth Pagadala[4,7], Layla Saidi[1], Catherine Z. Chen [2], Peng Gao[2], Miao Xu[2], Ivan Pavlinov[2], Bing Li[2], Wenwei Huang [2], Liqiang Chen[5], Jian Liu[4], Hang Xie [3], Wei Zheng [2] & Yihong Ye [1]✉

SARS-CoV-2 infection causes spike-dependent fusion of infected cells with ACE2 positive neighboring cells, generating multi-nuclear syncytia that are often associated with severe COVID. To better elucidate the mechanism of spike-induced syncytium formation, we combine chemical genetics with 4D confocal imaging to establish the cell surface heparan sulfate (HS) as a critical stimulator for spike-induced cell-cell fusion. We show that HS binds spike and promotes spike-induced ACE2 clustering, forming synapse-like cell-cell contacts that facilitate fusion pore formation between ACE2-expresing and spike-transfected human cells. Chemical or genetic inhibition of HS mitigates ACE2 clustering, and thus, syncytium formation, whereas in a cell-free system comprising purified HS and lipid-anchored ACE2, HS stimulates ACE2 clustering directly in the presence of spike. Furthermore, HS-stimulated syncytium formation and receptor clustering require a conserved ACE2 linker distal from the spike-binding site. Importantly, the cell fusion-boosting function of HS can be targeted by an investigational HS-binding drug, which reduces syncytium formation in vitro and viral infection in mice. Thus, HS, as a host factor exploited by SARS-CoV-2 to facilitate receptor clustering and a stimulator of infection-associated syncytium formation, may be a promising therapeutic target for severe COVID.

SARS-CoV-2, a single-stranded RNA virus coated with a membrane derived from host cells, injects its genetic materials into the host cytoplasm at the cell surface or following receptor-mediated endocytosis[1,2]. Both entry routes require the fusion of viral membranes with host membranes, which depends on the cell surface receptor ACE2 and the viral glycoprotein spike[3], a single-spanning homo-trimeric membrane protein[4]. During viral biogenesis, the spike undergoes a furin-mediated cleavage at the S1 site, generating two fragments, S1 and S2[5]. The S1 fragment contains the receptor binding domain (RBD) that exists in at least two conformations: In the up conformation, the spike binds ACE2 with high affinity[6], inducing TMPRSS2-mediated cleavage of the S2 fragment that might ultimately

[1]Laboratory of Molecular Biology, National Institute of Diabetes, Digestive, and Kidney Diseases, National Institutes of Health, Bethesda, MD 20892, USA. [2]The National Center for Advancing Translational Sciences, National Institutes of Health, Bethesda, MD 20850, USA. [3]Laboratory of Pediatric and Respiratory Virus Diseases, Division of Viral Products, Office of Vaccines Research and Review, Center for Biologics Evaluation and Research, Food and Drug Administration, Silver Spring, MD 20993, USA. [4]Eshelman School of Pharmacy, University of North Carolina, Chapel Hill, NC 27599, USA. [5]Center for Drug Design, College of Pharmacy, University of Minnesota, Minneapolis, MN 55455, USA. [6]Electron Microscopy Core, National Heart, Lung and Blood Institute, National Institutes of Health, Bethesda, MD 20892, USA. [7]Present address: Glycan Therapeutics Corp, 617 Hutton Street, Raleigh, NC 27606, USA. ✉e-mail: yihongy@mail.nih.gov

collapse an S2 fusion intermediate into a helical bundle to drive the fusion of viral membranes with the plasma membrane[5,7–9]. The constant emergence of new viral variants with improved spike cleavage and thus enhanced membrane fusion activity underscores TMPRSS2-mediated cleavage as a rate limiting factor in spike-induced membrane fusion[10]. The binding of ACE2 to spike also induces receptor-mediated endocytosis, transferring viral particles to late endosomes/lysosomes where a lysosomal protease can activate the spike similarly as TMPRSSP2 to promote membrane fusion[2,7,11–14].

Intriguingly, in addition to ACE2, recent studies have established the cell surface heparan sulfate (HS) as a critical co-receptor that assists ACE2 in viral entry[11,15–18]. HS refers to a class of negative charge-enriched polysaccharides attached to the specific membrane and secretory proteins collectively termed heparan sulfate proteoglycans (HSPG)[19]. Many viruses are thought to attach to the cell surface first by binding to HSPG[20]. For SARS-CoV-2, recent studies showed that HS could bind spike directly, forming an ACE2-containing ternary complex to promote SARS-CoV-2 endocytosis[11,15,17,21,22]. The role of HS in SARS-CoV-2 infection is critical in cells with low levels of ACE2 expression[16,23,24]. Accordingly, endocytosis-mediated SARS-CoV-2 cell entry can be inhibited by HS-binding drugs or HS mimetic compounds[11,15,25].

While ACE2-mediated viral endocytosis has been extensively studied, the mechanism underlying spike-induced membrane fusion needs to be better characterized. The spike-ACE2 interactions induce the fusion of the viral membranes with host membranes and also cause virus-infected cells to fuse with ACE2-positive neighboring cells[26]. The latter requires cell surface-displayed spikes, which likely originate from two distinct sources: 1) the fusion of the viral membrane with the plasma membrane and 2) de novo spike synthesis after viral entry. Strikingly, when the virus fuses with the plasma membrane, the viral membrane becomes part of the cell membrane[9], leading to an expansion of the spike-containing membranes and a dilution of the spike. Once the virus is inside the cell, de novo synthesis of virally encoded proteins begins, but the presence of an endoplasmic reticulum (ER)-retention signal should prevent most de novo synthesized spike from reaching the cell surface[27]. For these reasons, the level of surface-localized spike on infected cells is expected to be low. Nevertheless, large syncytia were observed in damaged lungs from postmortem COVID-19 patients[28–30] and are thought to promote SARS-CoV-2 spreading, evasion of antibody neutralization, and disease severity[31,32]. How spike maintains high fusogenic activities in infected cells remains unclear.

Our study reveals an unexpected function for the cell surface HS in SARS-CoV-2-induced syncytium formation. We show that upon spike ligation to ACE2, HS promotes the assembly of ACE2 into super-clusters to facilitate synaptogenesis, and thus, spike-induced cell-cell fusion. We further show that ACE2 clustering effectively concentrates the spike at fusion sites, which depends on a conserved linker in ACE2. Importantly, ACE2 clustering and syncytium formation can be targeted by an investigational drug that binds HS via specific sulfate groups, suggesting HS as a potential therapeutic target for alleviating severe COVID-19 symptoms associated with syncytium formation.

## Results

### A HS inhibitor mitigates SARS-CoV-2 infection in vitro and in a mouse model

Mitoxantrone (MTAN), an HS-binding drug, inhibits endocytosis-mediated entry of SARS-CoV-2 and other HS-dependent cargos[11,33]. However, MTAN's cytotoxicity has limited its use as a potential antiviral drug. To overcome this problem, we synthesized a collection of derivatives (Fig. 1a) and measured their cytotoxicity and antiviral activity against two pseudoviruses coated with the spike of different SARS-CoV-2 variants. MTAN contains a di-hydroxy-anthraquinone core and two symmetric arms bearing a hydroxyethyl amino group.

Increasing the arm length reduced both the toxicity and antiviral activity (Fig. 1b, c) while replacing the hydroxyethyl amino group with hydrogen did not affect either. Thus, modulating the arms cannot segregate the antiviral activity from cytotoxicity. By contrast, substituting several functional groups in the anthraquinone core reduced the cytotoxicity by at least one magnitude while only had a modest impact on the antiviral activity (LC1540 and LC1541) (Fig. 1b, c). We chose LC1541 (also known as Pixantrone or PIXN) for further study because of its potent inhibition on spike-bearing pseudoviruses (Supplementary Fig. 1a), strong affinity to HS (Supplementary Fig. 1b), and most importantly, its improved safety profile, as demonstrated in naïve mice (Supplementary Fig. 1c) and in patients[34].

Using a highly permissive Vero E6-derived cell line Vero TA6 (Vero E6 overexpressing human TMPRSS2 and ACE2), we found that an authentic SARS-CoV-2 variant (USA-WA1/2020) mainly accumulated in a LAMP1 positive lysosomal compartment after infection (Fig. 1d, e), consistent with endocytosis being the major entry path for this variant[13,24]. In PIXN-treated cells, the lysosomal accumulation of the spike-containing virus was significantly mitigated (Fig. 1d), suggesting that the drug inhibits viral entry at a step upstream of endocytosis. Furthermore, in a 3D-EpiAirway system simulating human airway infection (Fig. 1f)[35], PIXN-treated samples at both 24- and 96-h post infection (hpi) showed a reduction in viral load even at the lowest dose when compared to vehicle-treated controls (Fig. 1g). Meanwhile, using the release of lactate dehydrogenase (LDH) as an indicator of cytotoxicity, we detected no significant cell death in PIXN-treated samples except in prolonged treatment with a high dose (20 μM) (Fig. 1h). These results show that PIXN has a favorable anti-SARS-CoV-2 activity in vitro, further supporting the role of HS in SARS-CoV-2 cell entry.

To test whether HS affects SARS-CoV-2 infection in vivo, we evaluated the anti-SARS-CoV-2 activity of PIXN in a mouse model. To this end, we injected PIXN intravenously at two doses (50 μg/kg and 100 μg/kg) into K18-hACE2 transgenic mice expressing human ACE2[36]. We then infected the mice with live USA-WA1/2020. 72 h post-infection, we collected lung tissues and measured the viral load by qRT-PCR (Supplementary Fig. 1d). We found that for animals treated with PIXN at 100 μg/kg, the viral load was reduced by ~60% compared to the control group, while at a lower dose, a minor reduction was observed (Supplementary Fig. 1e). These results show that PIXN, as an HS-binding drug, can mitigate SARS-CoV-2 infection in vivo, although the activity is weak. The low in vivo anti-SARS-CoV-2 activity may be due to PIXN binding to HS in non-targeting tissues, which would reduce its effective concentration in the lung, suggesting that further optimizations are required to advance HS inhibitors into clinical trials as an antiviral agent.

### PIXN and MTAN interact with HS via specific sulfate groups

We next used a synthetic HS hexasaccharide (the six sugar moieties are designated as A−F, respectively) with defined sulfate groups (named 6-mer-NS2S3S6S) to characterize how PIXN and MTAN interact with HS (Fig. 2a) because the HS 6-mer binds these drugs similarly as longer heterogenous HS chains (see below). We first used NMR to measure chemical shift perturbations of HS 6-mer NS2S3S6S caused by PIXN. MTAN was omitted from this experiment because of its self-aggregation property at the concentration required for the NMR study. The result showed that most spectral changes happened on the sugar moieties B−D, centering around the C-3 position in N-sulfo-D-glucosamine B and the anomeric position in glucuronic acid C (Fig. 2b, c; Supplementary Fig. 2a; Supplementary Table 1). Reciprocal titration of HS 6-mer NS2S3S6S to PIXN showed that the two -NH groups on the arms were most significantly affected (Fig. 2d). These results suggest that HS might use specific sulfate groups (e.g., 2S, NS, and 6S) in the sugar moieties B−D to interact with the -NH groups of PIXN. This model is consistent with the observation that the -NH groups are conserved in MTAN.

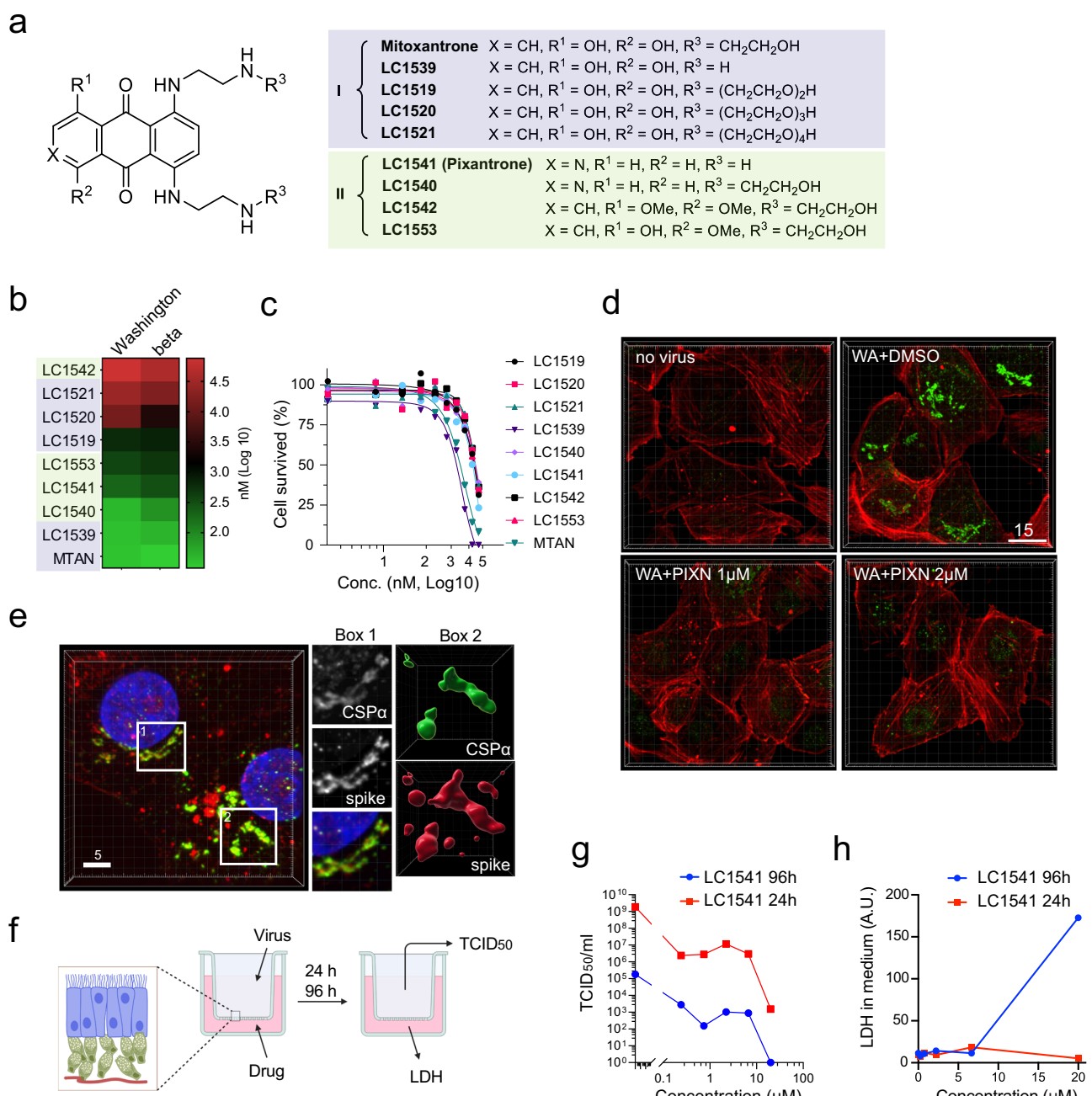

**Fig. 1 | A MTAN-related drug inhibits SARS-CoV-2 infection. a** The structure of MTAN and derivatives. **b** The inhibitory activity of MTAN derivatives on the entry of pseudoviral particles coated with the spike of the indicated variants. Shown is a heat map of IC$_{50}$ averaged from three repeats. **c** The effect of MTAN derivatives on cell viability. HEK293T cells were treated with the indicated chemicals for 48 h before cytotoxicity was measured. The graph represents means of two biological repeats. **d** PIXN inhibits the entry of USA-WA1/2020. Vero TA6 cells were infected with live SARS-CoV-2 USA-WA1/2020 strain (WA) or mock-infected for 4 h in the presence of the indicated compounds. Cells were fixed and stained with a spike antibody in green and an actin dye in red. Scale bar, 15 μm. Images are representative of three biological repeats. **e** The USA-WA1/2020 variant enters cells mostly via endocytosis. Vera TA6 cells were infected with the USA-WA1/2020 for

4 h, fixed, and stained with antibodies against the lysosome-associated protein CSPα (green), the spike (red), and with Hoechst (blue). Shown are examples of cells in a 3D view (left). Scale bar, 5 μm. The Box1-labeled area is enlarged in the middle panels. The right panels show a surface-rendered view of the Box2 area. **f** A schematic illustration of the 3D-EpiAirway assay. Created by BioRender.com. **g** PIXN inhibits SARS-CoV-2 infection in the 3D EpiAirway model. TCID$_{50}$ was determined either 24 h or 96 h after the tissues were treated with PIXN at the indicated concentrations and then air-infected with USA-WA1/2020 at a MOI of 0.1 for 1 h. The curves represent means of two biological repeats. **h** PIXN does not induce significant cytotoxicity in the 3D EpiAirway model. A.U., arbitrary unit. The curves represent means of two biological repeats. Source data are provided as a Source Data file.

To validate the NMR results, we used a binding assay based on the observation that both MTAN and PIXN absorb light at 650 nm[11] but the interaction of these drugs with HS or HS 6-mer NS2S3S6S reduces Ab650 (Fig. 2e; Supplementary Fig. 2b). Plotting the change in Ab650 over HS concentrations suggested that PIXN bound to 6-mer-NS2S3S6S with a similar affinity as MTAN. However, the maximum binding for PIXN was about two-fold higher than that of MTAN (Fig. 2f). As expected, an HS 6-mer analog bearing no sulfate group (6-mer-0S) did not interact with either PIXN or MTAN (Fig. 2g; Supplementary Fig. 2c), demonstrating a sulfate-dependent interaction with these drugs.

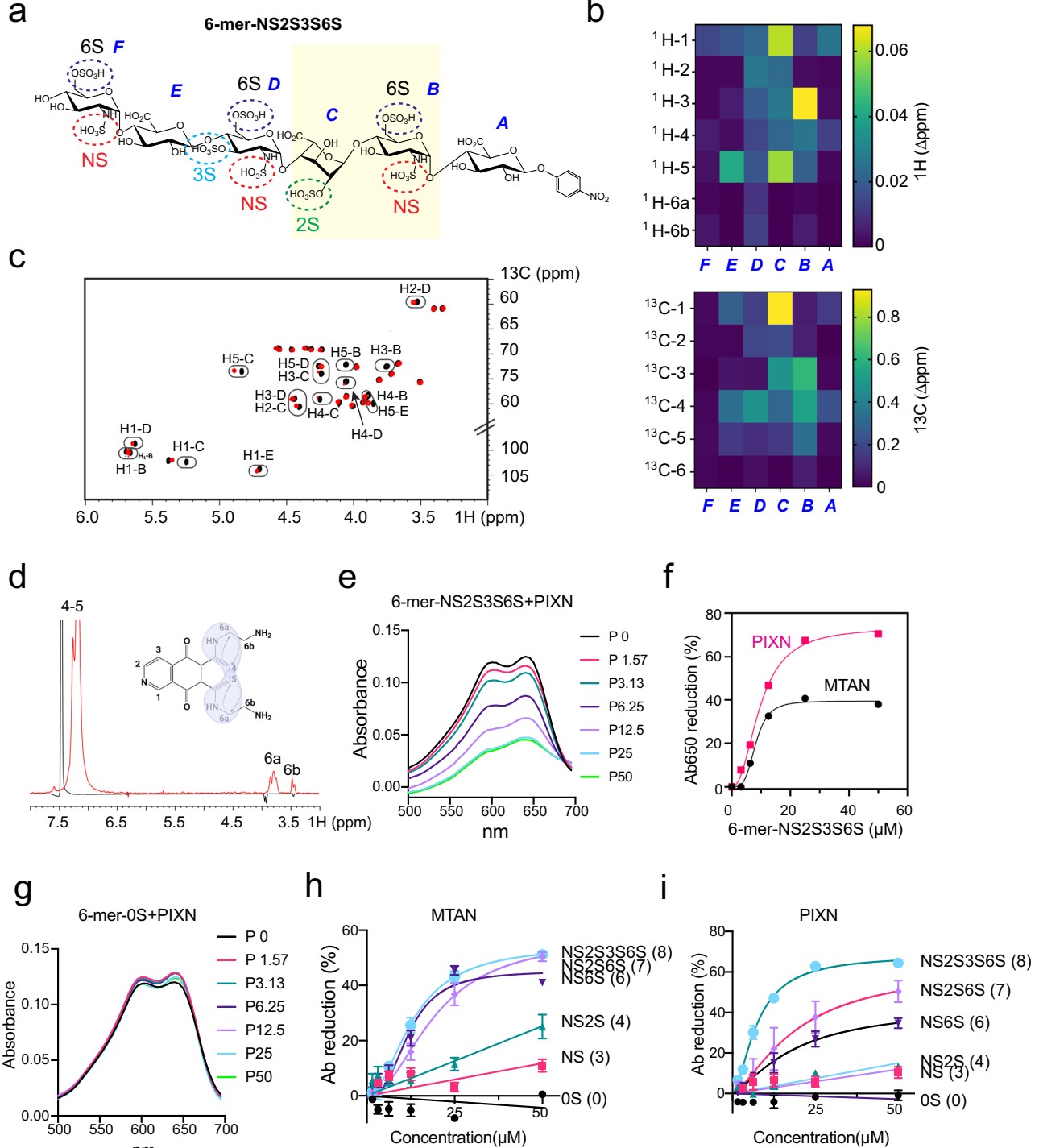

**Fig. 2 | PIXN and MTAN bind distinct sulfate groups on HS. a** The chemical structure of 6-mer NS2S3S6S. This hexasaccharide bears three N-sulfate, three 6-sulfate, one 2S, and one 3S groups. The yellow shaded domain is most significantly affected by PIXN as determined by NMR. **b** Heat maps showing the 6-merNS2S3S6S chemical shift perturbation (CSP) caused by PIXN binding. Absolute values of Δppm for both $^{1}H$ (top) and $^{13}C$ (bottom) were plotted. The central part of 6-merNS2S3S6S shows higher valus of CSP, suggesting that the interaction takes place only with the inner residues of the oligosaccharide. **c** $^{1}H$, $^{13}C$ HSQC spectra of the 6-merNS2S3S6S. The figure shows a superposition of HSQC spectra of the free substrate in black and with addition of the ligand PIXN in red (2:1 ratio of 6-mer:PIXN). Peaks with notable Chemical Shift Perturbation (CSP) are circled and labeled. **d** 1D-NOE comparison of free PIXN (black) and upon binding (red) suggests a modification in the spatial proximity of protons surrounding the aromatic amino groups when the interaction with 6-merNS2S3S6S occurs. **e** 6-mer NS2S3S6S binding changes the absorption spectra of PIXN. The numbers indicate drug concentrations (μM). **f** Both PIXN and MTAN bind HS 6-mer NS2S3S6S. PIXN or MTAN (50 μM) was incubated with 6-mer NS2S3S6S at the indicated concentrations. The changes in $Ab_{650}$ were plotted. **g** As in (**e**) except that a HS 6-mer without any sulfate groups was used. **h**, **i** The differential contribution of the sulfate groups to MTAN and PIXN binding. As in (**f**), except that HS 6-mer analogs bearing different number of sulfate groups were used. Error bars indicate means ± s.e.m. $n = 3$ biological repeats. Source data are provided as a Source Data file.

To identify the sulfate group(s) involved in drug binding, we performed a binding study using synthetic HS 6-mers containing only given types of sulfate groups (Supplementary Fig. 2d). The results showed that the drug binding depends on both sulfate number and position (Fig. 2h, i); Low binding was generally detected for HS 6-mer carrying three NS groups. For MTAN, the addition of three 6S groups (NS6S) increased the affinity dramatically, while having additional 2S and 3S groups (NS6S2S or NS2S3S6S) did not further improve MTAN binding (Fig. 2h). By contrast, 6-mer NS6S only had a modestly increased affinity for PIXN compared to 6-mer NS, while adding 2S and 3S maximized the affinity to PIXN (Fig. 2i). Thus, while PIXN and MTAN both bind HS via sulfate groups, they have different preferences for the sulfate position.

## Pharmacological inhibition of HS mitigates SARS-CoV-2-induced cell-cell fusion

Having established PIXN and MTAN as HS-binding drugs (Fig. 2) that inhibits endocytosis-mediated entry of SARS-CoV-2 (Fig. 1d)[37], we tested whether these drugs inhibited SARS-CoV-2 entry via fusion at the plasma membrane. We chose the Delta variant (B.1.617.2) because recent studies suggested that this variant has the most robust membrane fusion-stimulating activity[10], and thus is thought to enter permissive cells primarily via the plasma membrane[38]. Consistent with this model, when Vero TA6 cells infected with the Delta variant were stained with an antibody against the SARS-CoV-2 nucelocapsid protein (NP) at 4 hpi, we detected strong NP signals as small puncta throughout the cytoplasm, but no lysosomal accumulation of the virus was observed (Supplementary Fig. 3a). Cells treated with either PIXN or MTAN during infection still contained NP-positive puncta, but the intensity was reduced by ~50% (Fig. 3a, b). Thus, both PIXN and MTAN inhibit Delta entry via the plasma membrane.

As expected from the robust fusogenic activity of the Delta spike, we observed many multinuclear syncytia in cells at 4 hpi with the Delta variant (Fig. 3c, d). Since de novo spike synthesis did not occur at this point (Supplementary Fig. 3b–d), the syncytium formation must be caused by the cell surface spike acquired during viral entry, as seen by staining cells with spike antibodies (Supplementary Video 1). Consistent with this notion, infection with the endosome-routed USA-WA1/2020 only generated syncytia after prolonged infection, which produced cell surface spike by de novo synthesis (Supplementary Fig. 3a, b). Syncytium formation was completely dependent on infection because no syncytium was detected in uninfected cells (Supplementary Fig. 3a, b); Only a few syncytia comprising up to three nuclei could be seen when a low MOI (0.01) was used (Supplementary Fig. 3c, d). By contrast, in samples infected at MOI of 0.5, some giant syncytia containing up to 80 nuclei were seen (Fig. 3c). Because our experimental conditions did not support "fusion-from-without", a model of membrane fusion induced by high concentrations of viral particles without presenting spike to the cell surface[39], these giant syncytia must be formed by multiple rounds of cell-cell fusion. It is intriguing that the limited spike molecules transferred from the viral particles can maintain such high fusogenic activity, particularly as the membrane expansion during viral entry significantly dilutes the spike concentration.

Interestingly, when cells infected with the Delta variant were treated with PIXN or MTAN, the syncytium size was much reduced, with most syncytia containing only 4-6 nuclei (Fig. 3c, d). These findings raise the possibility that HS might enhance the fusogenic activity of the spike on the cell surface, which PIXN and MTAN antagonize.

## HS promotes spike-induced syncytium formation

To elucidate the role of HS in spike-induced cell−cell fusion, we optimized a co-culture-based fusion assay (Fig. 3e). We co-transfected 293 T cells with a Delta spike- and a mCherry-expressing plasmids (Effector). To overcome the ER retention of the spike, we deleted the C-terminal tail of the spike, which enhances its trafficking to the cell surface[40]. Transfected cells were added in suspension to a monolayer of 293 T cells stably expressing ACE2-GFP (Acceptor). Spike-expressing cells were round and mostly mCherry-positive, but after fusing with ACE2 cells, they attached to the surface and flattened into irregular shapes. After incubation, unfused cells were removed by washing. The fusion efficiency could be monitored by counting the number of nuclei per syncytium under fluorescence microscopy. In this assay, syncytium formation strictly depends on the spike in effector cells and ACE2 in acceptor cells[37]. Additionally, both PIXN and MTAN inhibited syncytium formation in this assay (Supplementary Fig. 3e, f)[37], suggesting that the assay recapitulates infection-associated syncytium formation.

We next created HS-deficient ACE2-GFP cells by CRISPR-mediated knockout (KO) of *SLC35B2*, a Golgi-localized sulfate transporter essential for the sulfation of HS. Staining cells with a super-charged cyan fluorescent protein (CFP+) (Fig. 3f), a positive charge-bearing fluorescence protein that binds HS with high affinity, confirmed the HS deficiency in the KO cells[11]. When *SLC35B2* KO or control ACE2-GFP cells were co-cultured with spike/mCherry cells, syncytia formed by *SLC35B2* KO cells were significantly smaller than in wild-type (WT) control despite similar levels of ACE2-GFP expression (Fig. 3g, h). Notably, the cell fusion defect of *SLC35B2* deficient cells was rescued when heparin, a heavily sulfated HS, was added to the medium, attributing the phenotype to HS deficiency. Heparin treatment also increased the size of syncytia in WT cells (Fig. 3g, h). These results demonstrate a critical role for the cell surface HS in spike-mediated cell-cell fusion, which can be substituted by unanchored heparin.

To further validate the role of HS in membrane fusion, we treated ACE2-GFP cells with heparinase I/III to remove the cell surface HS (Fig. 3f). Like SCL35B2 KO cells, heparinase-treated ACE2-GFP cells, when co-cultured with spike/mCherry cells, produced syncytia of smaller sizes (Supplementary Fig. 3g, h). Likewise, heparinase treatment also significantly reduced the syncytium size of Vero TA6 cells infected with the Delta SARS-CoV-2 (Fig. 3i, j). Because removing HS from acceptor cells alone is sufficient to mitigate cell-cell fusion and because high concentrations of heparin are required to restore the fusion activity in *SLC35B2* KO cells, it appears that HS acts most effectively when positioned in cis to ACE2-containing membranes (see discussion).

## HS enables synapse-like cell−cell contacts in spike-mediated cell−cell fusion

To fuse with acceptor cells, an effector cell needs to go through the following steps conceptually: 1) cell-cell contact formation, 2) generating a small fusion pore, 3) the pore widening into a significant gap, and 4) complete fusion of the two cells (Fig. 4a). We used the co-culture assay to narrow down the step(s) affected by PIXN. Specifically, in addition to measuring the syncytium size (Fig. 4b), we counted the number of mononuclear cells with mCherry and ACE2-GFP (Fig. 4c). These cells are in a semi-fusion state (F3 in Fig. 4a) with a fusion pore forming, a state downstream of the hemifusion stage[41]. In vehicle-treated reactions, ~4% of ACE2-GFP cells were detected in this semi-fusion state after 30 min incubation with spike/mCherry cells. In contrast, in PIXN-treated conditions, the number of semi-fusion cells increased to ~9% (Fig. 4c; Supplementary Fig. 4a). After further incubation, semi-fusion cells were almost undetectable in the control reaction because most cells had completed at least one round of fusion. However, in the presence of PIXN, ~8% of ACE2-GFP cells remained in this state, suggesting that PIXN arrests the fusion reaction in the semi-fusion state.

To further elucidate the role of HS, we used 4D live cell confocal microscopy to monitor the effect of PIXN on cell−cell fusion in real-time. Analyses of randomly selected fusion events confirmed that PIXN treatment delayed both the entry of mCherry into ACE2-GFP cells and the appearance of a visible pore. However, once a visible pore was

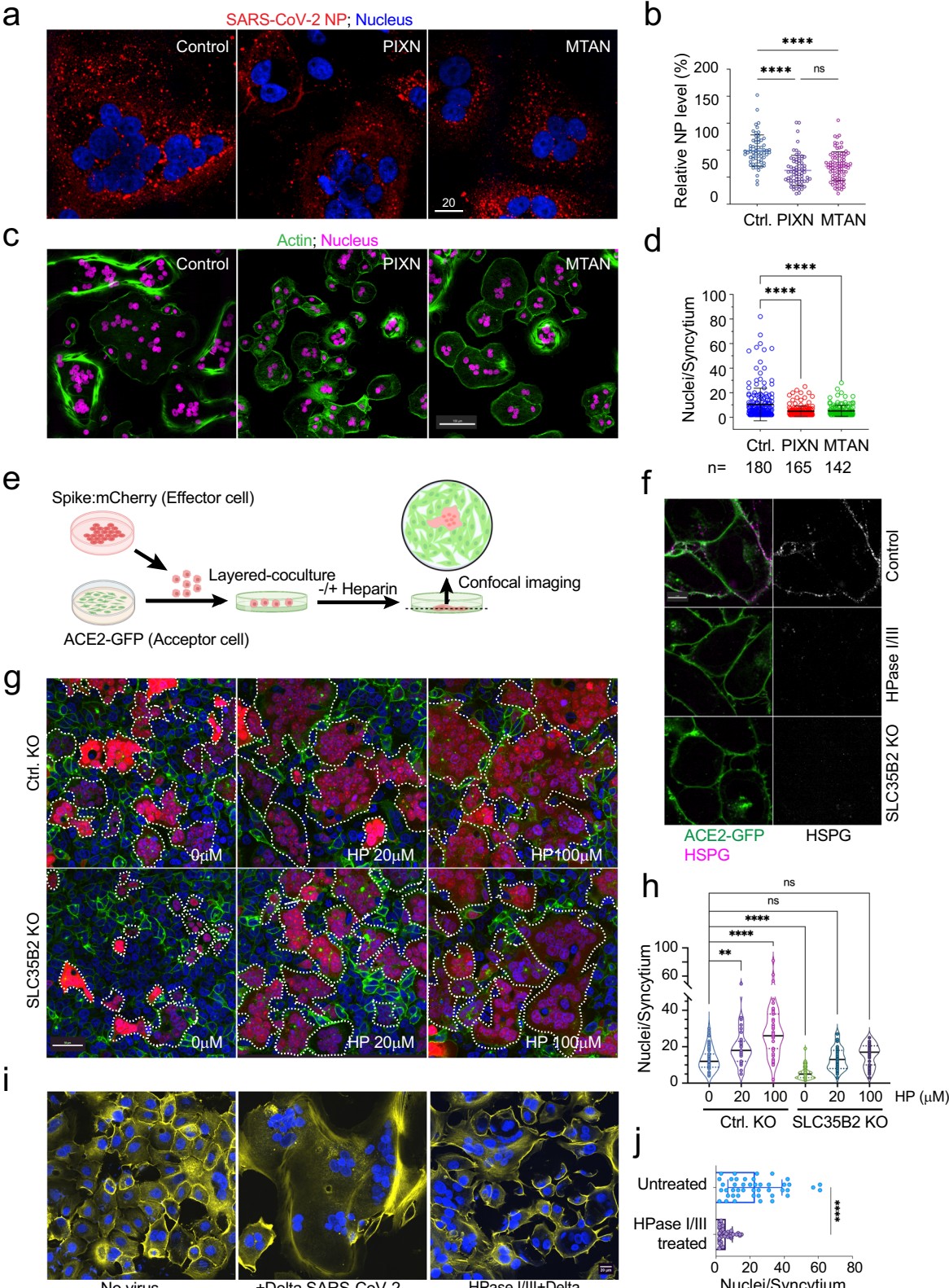

formed, the pore widening was unaffected (Fig. 4d, e). These results further suggest that HS facilitates fusion pore formation.

Since biochemical studies showed that neither PIXN nor MTAN inhibited HS binding to the spike and that PIXN did not affect the interaction of the spike with ACE2 in vitro regardless of whether HS was present (Supplementary Fig. 4b, c), HS is likely dispensable for the initiation of cell–cell contact. Indeed, when spike cells contact ACE2

cells, the spike is processed by the cell surface TMPRSS2, generating an S2′ fragment and a fusion peptide. This process was also unaffected by PIXN or knockout of *SLC35B2* in acceptor cells (Supplementary Fig. 4d, e).

Transmission electron microscopy showed that after mixing spike- and ACE2- cells, these cells form synapse-like cell–cell contacts characterized by long juxtaposed plasma membranes interrupted by

**Fig. 3 | The cell surface HS promotes spike-induced cell-cell fusion. a** Vero TA6 cells treated with the indicated drugs were infected with live Delta variant at MOI of 0.5 for 4 h. Cells were stained with an antibody against the viral nucleocapsid (NP) protein (red) and a Hoechst dye (blue). Scale bar, 20 μm. **b** Quantification of the NP signals in individual cells from four biological repeats. ****, $p < 0.0001$ by two-sided unpaired student's $t$ test, $n = 4$, ns, not significant. **c** Vero TA6 cells infected with live Delta variant at MOI of 0.5 in the absence (control) or presence of the indicated drugs for 4 h. Cells were stained by an actin dye (green) and Hoechst (magenta). **d** Quantification of the syncytium size in (**c**). The n values indicate the number of syncytium counted. Error bars indicate means ± SD, ****, $p < 0.0001$ by two-sided unpaired student's $t$ test. $n = 4$ biological repeats. **e** A schematic illustration of the optimized cell–cell fusion assay. Created by BioRender.com. **f** Super-charged CFP

staining validates the lack of cell surface HS by either heparinase I/III treatment or *SLC35B2* knockout (KO). Images are representative of three biological repeats. **g** WT (Ctrl.) and *SLC35B2* CRISPR knockout (KO) cells expressing ACE2-GFP were incubated with spike/mCherry cells in the absence or presence of heparin (HP) at the indicated concentrations and imaged after 1 h. Dashed lines mark examples of syncytia. Scale bar, 50 μm. **h** Quantification of the experiments represented in (**g**). **, $p < 0.006$; ****, $p < 0.0001$ by one-way ANOVA. $n = 3$ biological repeats. **i** Vero TA6 cells were mocked treated or treated with heparinase I/III for 3 h prior to infection with live SARS-CoV-2 Delta variant (MOI = 0.5). Cells were fixed and stained with an actin dye (yellow) and Hoechst (blue). **j** Quantification of (**i**). Error bars indicate means ± SD. ****, $p < 0.0001$ by two-sided unpaired student's $t$ test. $n = 3$ biological repeats. Source data are provided as a Source Data file.

bubble-like structures. The paralleled membranes, unseen when ACE2 cells were incubated with cells without spike (Supplementary Fig. 4f), are only separated by a gap of less than 20 nm (Fig. 4f, right panels; Fig. 4g). Importantly, we frequently observed obscure membrane boundaries at these membrane contact sites, probably caused by localized fusion events. By contrast, in PIXN-treated samples, cell-cell contacts were seen, but membranes were only held together at a few spots, leaving significant gaps in between, and fusion pores were rarely seen. Thus, HS facilitates the formation of tight membrane junctions between the effector and acceptor cells during spike-mediated cell–cell fusion.

## HS enhances ACE2 clustering at the fusion site

4D live cell confocal imaging revealed that within minutes following the contact of spike cells with ACE2-GFP cells, ACE2-GFP was rapidly concentrated at the cell-cell contact sites (Fig. 4d, Supplementary Video 2), consistent with a recent report[29]. The ACE2-GFP clusters expanded over time, forming synapse-like super assemblies with a diameter of 10–20 μm. Fluorescent recovery after photobleaching (FRAP) showed that ACE2-GFP was almost entirely immobile in these super-clusters (Fig. 5a, b), suggesting that these clusters are formed by ordered intermolecular interactions as opposed to liquid-liquid phase separation. As expected, immunostaining with spike antibodies also detected similar spike accumulation at the cell-cell contact sites (Fig. 5c). By contrast, in the absence of ACE2-GFP cells, the spike was uniformly distributed on the cell surface as small puncta (Supplementary Fig. 5a). Since recombinant spike fragments containing the RBD could completely block ACE2 cluster formation (Fig. 5d), ACE2 super-clustering requires spike-ACE2 interactions. Interestingly, 3D confocal imaging revealed enrichment of actin filaments around ACE2 super-clusters, resembling those in the immune synapse (Supplementary Fig. 5b). Because ACE2 clustering always precedes the entry of mCherry into the ACE2-GFP cell, we presumed that ACE2 clustering plays a role in spike-induced cell–cell fusion.

To determine whether HS regulates ACE2 cluster formation, we compared the size of ACE2 clusters in WT cells to that in the *SLC35B2* KO cells because the latter lack HS. When SCL35B2 KO ACE2-GFP cells were incubated with spike/mCherry cells, the size of ACE2 clusters was much smaller than those in WT ACE2-GFP cells (Fig. 5e, g). Likewise, PIXN treatment also significantly reduced the size of ACE2 clusters (Fig. 5f, h). Kinetic analysis of ACE2 cluster formation by 4D confocal imaging further confirmed a receptor clustering defect when the cell surface HS is inhibited by PIXN (Fig. 5i). These results indicate that HS facilitates ACE2 super-cluster formation, which promotes cell-cell fusion.

## HS acts via a conserved ACE2 linker to promote receptor clustering and cell–cell fusion

As HS does not change the affinity of the spike for ACE2, we postulated that HS might induce a conformational change in ACE2, exposing a motif that drives receptor clustering. ACE2 is a single-spanning membrane protein with a large extracellular domain (ECD) for spike and

ligand binding. The ECD is connected via an unstructured linker to the transmembrane domain, which mediates receptor dimerization. Intriguingly, albeit with no designated function, the unstructured linker is evolutionarily conserved (Fig. 6a).

To test whether the linker segment (LS) is involved in spike-induced cell-cell fusion, we created a 293 T cell line expressing a mutant ACE2-GFP with the LS replaced by a synthetic linker bearing a glycine-serine repeat of the same length (ACE2-GS). Immunoblotting and fluorescence imaging showed that ACE2-GS-GFP was expressed at a similar level as WT ACE2-GFP and localized to the cell surface and endocytic vesicles similarly to WT ACE2-GFP (Supplementary Fig. 6a, b). Furthermore, binding studies showed that ACE2-GS-GFP bound to the spike with a similar affinity as WT ACE2-GFP (Supplementary Fig. 6c). Nevertheless, when ACE2-GS-GFP cells were co-cultured with spike-expressing cells, spike super-clustering was significantly reduced (Supplementary Fig. 5a, b), so was ACE2-GFP super-clustering (Fig. 6b; Supplementary Fig. 6d). Unlike in WT ACE2-GFP cells, PIXN treatment in ACE2-GS-GFP cells did not further reduce the size of ACE2 clusters (Fig. 6b). These results suggest that the linker domain is critical for HS-assisted ACE2 super-cluster formation, which in turn concentrates the spike at cell-cell contact sites.

Consistent with the strong correlation between ACE2 clustering and syncytium formation, the ACE2-GS-GFP cells had dramatically reduced fusion activity, resulting in syncytia of reduced size similar to PIXN-treated WT ACE2-GFP cells (Fig. 6c, d). Moreover, while heparin stimulated the fusion of ACE2-GFP cells with spike cells, it failed to do so in ACE2-GS-GFP cells co-cultured with spike cells (Fig. 6e). These observations suggest HS acts via the ACE2 LS domain to promote ACE2 super-clustering and cell-cell fusion. By contrast, the entry of spike-coated pseudoviruses was unaffected in ACE2-GS-GFP cells compared to WT ACE2-GFP cells (Supplementary Fig. 6e). Thus, ACE2 super-cluster formation appears dispensable for endocytosis-mediated viral entry.

## HS enhances spike-mediated ACE2 clustering in a cell-free system

We developed a cell-free receptor clustering assay to test whether HS directly promotes ACE2 clustering (Fig. 7a). Because purified ACE2 did not cluster in the presence of spike and HS in solution, we attached Alexa$_{565}$-labeled ACE2$_{1-740}$ to a cover glass coated with a lipid bilayer that consisted of 1-palmitoyl-2-oleoyl-sn-glycero-3-phospholcholine (POPC), 1,2-dioleoyl-sn–glycero-3-phosphoethanolamine-N-[methoxy(polyethylene glycol)-5000] (PEG-5000 PE), and 1, 2-dioleoyl-sn-glycero-3-[(N-(5-amino-1-carboxypentyl)iminodiacetic acid) succinyl] (DOGS-NTA) via a C-terminal 6xHis tag. This protein contains the ACE2 ECD and the conserved LS. In our method, the protein is anchored to the lipid bilayer in a highly mobile state, and importantly, with a topology mimicking membrane-anchored full-length ACE2. When membrane-anchored ACE2$_{1-740}$ was incubated with recombinant spike trimer, round-shaped bright speckles were formed due to ACE2 clustering, while incubating with buffer, the RBD domain of the spike, or HS alone did not induce ACE2 clustering (Fig. 7b, c;

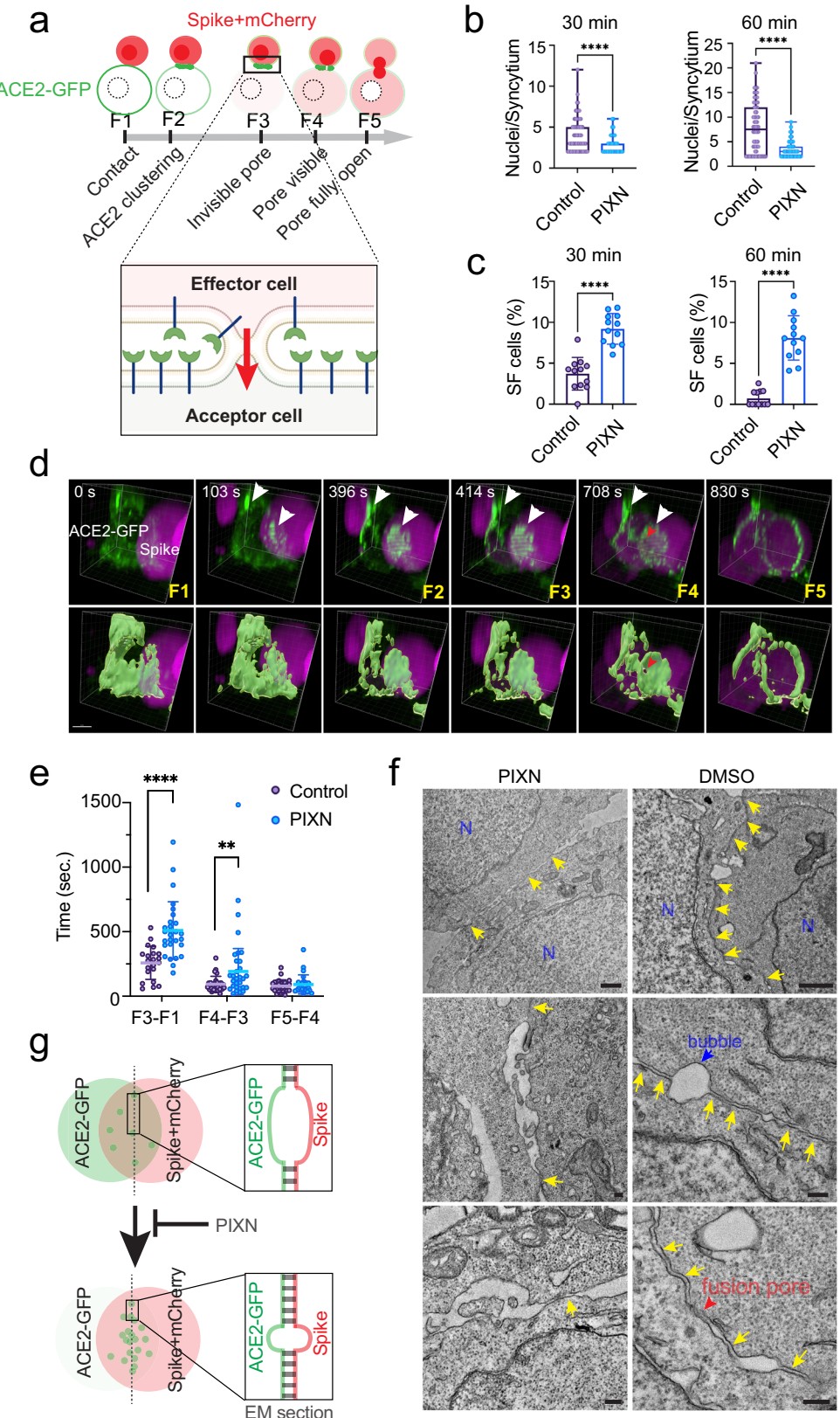

Supplementary Fig. 7a). Consistent with our model, ACE2 clustering with spike alone was inefficient, but when HS and spike were added together, ACE2 cluster formation was significantly accelerated. Like ACE2 clusters in cells, ACE2 in in vitro-assembled clusters was also immobile, as demonstrated by FRAP (Fig. 7d). These results suggest that HS directly promotes ACE2 clustering in the presence of the spike.

Surprisingly, adding PIXN did not abolish spike/HS-induced ACE2 clustering. However, under this condition, ACE2 clustering generated mostly elongated filaments (Supplementary Fig. 7b), suggesting that PIXN alters the mode of ACE2 self-assembly. The differential effect of PIXN on ACE2 clustering in cells vs. in vitro suggests that additional factors modulate ACE2 self-interaction on the

**Fig. 4 | HS promotes fusion pore formation to enhance spike-induced cell–cell fusion. a** A schematic illustration of the cell–cell fusion process. F1–F5 indicated different fusion stages identified by 4D imaging. Created by BioRender.com. Spike-induced cell-cell fusion was performed in the absence or presence of 1 μM PIXN for 30- or 60-min. The number of nuclei per syncytium (**b**) and semi-fusion (SF) cells (**c**) in randomly selected fields were counted. n = 80 (Control) and 66 (PIXN) syncytia for 30 min and 67 (Control) and 75 (PIXN) for 60 min from two biological repeats. The box graphs show the interquatile values (50%) by bound boxes, the maxima, the minima and the medians by lines. Representative images are presented in Supplementary Fig. 4a. Error bars in (**c**) indicate means ± SD, n = 12 fields/condition from two biological repeats. ****, p < 0.0001 by two-sided unpaired student's t test. **d** 4D confocal imaging of spike-induced cell–cell fusion. The bottom panels show surface-rendered ACE2-GFP signals. White arrowheads indicate ACE2-GFP clusters. The red arrowhead labels a visible fusion pore. Scale bar, 5 μm. **e** PIXN arrests the fusion at the semi-fusion stage and delays the appearance of visible fusion pores. The graph shows the times between the indicated fusion stages measured by 4D live cell imaging. Error bars indicate means ± SD, **, p < 0.008, ****, p < 0.0001 by two-sided unpaired student's t test. **f** PIXN disrupts the synapse-like cell-cell contacts between spike and ACE2-GFP cells. Shown are representative EM images of co-cultured cells untreated or treated with PIXN. N, nucleus; arrows show juxtaposed membranes at the contact sites. The red arrow shows an example of disrupted membranes possibly caused by fusion. Scale bars, 1 μm for top panels, 200 nm for other panels. Images are representative of two biological repeats. **g** A schematic diagram of the result in (**f**). Source data are provided as a Source Data file.

cell surface, preventing filament-like ACE2 clusters in the presence of PIXN.

## Discussion

The cause of clinical heterogeneity among SARS-CoV-2-infected patients is unclear, but it is suggested that spike-induced cell-cell fusion may be a contributor to inflammation, thrombosis, or lymphopenia observed in severe COVID-19 patients because multi-nuclei-containing syncytia were often seen in damaged lung tissues from posthumous COVID-19 organs[28–30], and because spike-induced syncytia can rapidly internalize lymphocytes, resulting in a unique cell-in-cell structure seen in COVID-19 tissues but not in other types of pneumonia[28]. When expressed in neurons, spike can even drive the fusion of neurons with neurons or glia, which might contribute to the neurologic symptoms associated with long COVID-19[42]. Furthermore, spike-induced cell-cell fusion may contribute to viral transmission. Since viruses do not need to exit the infected cells, it was proposed that this transmission route allows the virus to escape immune surveillance and antibody-mediated neutralization[31,32]. Intriguingly, SARS-CoV-2 variants of concern have different fusogenic potencies with the Delta strain having the highest membrane-fusion inducing activities. Coincidentally, studies have associated the emergence of the Delta variant with an increased risk of COVID-19-related hospitalization[43,44].

In infected cells, the surface spike concentration is usually low. Nevertheless, spike-positive cells can effectively fuse with effector cells expressing ACE2 without de novo synthesis of the spike (Fig. 3; Supplementary Fig. 3). How can the spike be so efficient at inducing membrane fusion? Given the fast fusion reaction, new protein synthesis is likely unable to compensate for the rapid reduction of the surface spike, particularly given the ER retention signal in the spike. Our data support a model in which HS facilitates an allosteric conformational change in spike-bound ACE2, allowing a conserved linker to promote ACE2 clustering. Since clustered ACE2 can engage the spike on the opposite membrane, these multivalent protein interactions also concentrate the spike (Fig. 5c) while bringing the effector and acceptor membranes together to form a synapse-like structure in preparation for membrane fusion. This model would explain the potent fusogenic activity of the spike despite the low concentration at the cell surface (Fig. 7e). How HS facilitates ACE2 super-cluster formation remains to be determined. Noticeably, it was previously shown that HS could enhance FGF signaling by promoting FGF2 receptor dimerization upon ligand binding[45]. It was proposed that HS might bind two receptor dimers simultaneously to link these receptors together, a mechanism that might contribute to ACE2 oligomerization (Fig. 7e). However, other alternative models cannot be excluded, particularly given the recent finding that ACE2 might function as a monomer during spike-mediated viral entry[46].

Intriguingly, our results suggest that HS functions predominantly on the ACE2-containing membrane because HS on spike-containing membranes fails to compensate for the loss of HS on the ACE2 cells.

Thus, HS might interact with the spike via a specific configuration not achievable when positioned in cis to the spike. Furthermore, free HS can promote ACE2 clustering and membrane fusion when added to the medium at high concentrations, suggesting that membrane association may increase local HS concentration and thus enhance its affinity to the spike. Consistent with this idea, HSPG antibody staining did detect HS in small puncta on the cell surface[11].

Our study establishes PIXN and MTAN as HS-binding drugs targeting specific sulfate groups in HS. Although both drugs were initially reported as DNA topoisomerase inhibitors[47] and had been evaluated extensively in clinics to treat various tumors[48,49], PIXN has an overall improved clinical safety profile[34]. This fact, combined with the observed in vivo anti-SARS-CoV-2 activity, suggests further testing the anti-viral activity of PIXN or other HS inhibitors in clinical trials for patients with severe COVID-19 symptoms.

Spike-induced membrane contacts are morphologically similar to previously reported immune synapses. Both immune and spike-induced synapses are enriched in actin filaments, organized into ring-shaped structures surrounding the juxtaposed membranes. The dynamic movement of actin filaments may provide a mechanical force that drives the dilation of the fusion pore during cell-cell fusion. Interestingly, in T cell receptor (TCR)-mediated immune synapses, ligand-receptor interaction is accompanied by the formation of large TCR clusters analogous to the ACE2 clusters observed in this study[50]. Thus, an intriguing question is whether HS may play a more general role in shaping the protein interaction networks at the cell surface during cell-cell communications and cell signaling. Indeed, HS was reported to regulate the activation of many signaling receptors, including the FGF receptor, Wnt, and Hedgehog receptor[51]. Whether these receptors form super-clusters in an HS-regulated manner upon ligand binding awaits future investigations.

## Methods
### Chemical synthesis

**General procedures.** All commercial reagents were used as provided unless otherwise indicated. An anhydrous solvent dispensing system (J. C. Meyer) using two packed columns of neutral alumina was used for drying THF, Et$_2$O, and CH$_2$Cl$_2$, whereas two packed columns of molecular sieves were used to dry DMF. Solvents were dispensed under argon. Flash chromatography was performed with RediSep R$_f$ silica gel columns on a Teledyne ISCO CombiFlash® R$_f$ system using the solvents as indicated. Nuclear magnetic resonance spectra were recorded on a Varian 600 MHz or Bruker 400 MHz spectrometer with Me$_4$Si or signals from residual solvent as the internal standard for $^1$H or $^{13}$C. Chemical shifts are reported in ppm, and signals are described as s (singlet), d (doublet), t (triplet), q (quartet), m (multiplet), and br s (broad singlet). Values given for coupling constants are first order. High resolution mass spectra were recorded on an Agilent TOF II TOF/MS instrument equipped with either an ESI or APCI interface at the Center for Drug Design, University of Minnesota (Minneapolis, MN, USA).

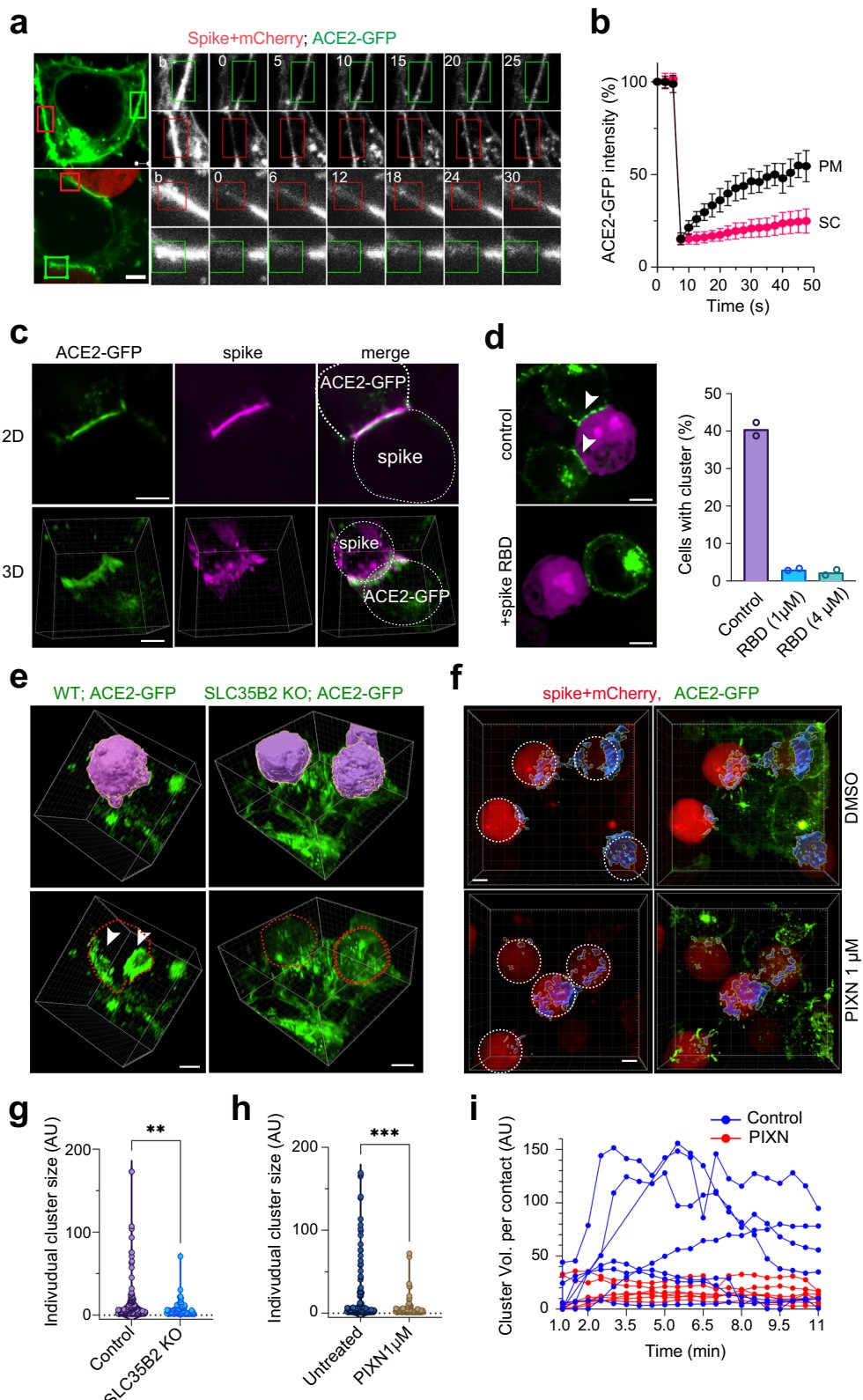

**LC-1541 (PIXN).** To compound 6,9-difluorobenzo[g]isoquinoline-5,10-dione (220 mg, 0.89 mmol, 1.0 eq.) in THF (20 mL) under argon atmosphere was added ethane-1,2-diamine (0.6 mL, 8.97 mmol, 10.0 eq.). The resultant solution was stirred at 50 °C for 24 h. The mixture was diluted with methanol (10 mL) and the solvents were removed under vacuum. The crude product was purified by column chromatography (30% MeOH: NH$_4$OH (9:1) in

CH$_2$Cl$_2$) to give compound LC-1541 as a blue gum (95 mg, 56%). $^1$H NMR (600 MHz, CD$_3$OD) δ 9.44 (s, 1H), 8.88 (d, $J$ = 5.2 Hz, 1H), 8.14 (d, $J$ = 5.2 Hz, 1H), 7.52 (d, $J$ = 4.2 Hz, 2H), 3.71 (td, $J$ = 6.4, 2.0 Hz, 4H), 3.13 (q, $J$ = 6.0 Hz, 4H). HRMS (ESI$^+$): m/z calcd for C$_{17}$H$_{20}$N$_5$O$_2$ [M + H]$^+$ 326.1612, found 326.1602. The following compounds were prepared in a fashion similar to the one described for LC-1541.

**Fig. 5 | HS facilitates spike-induced ACE2 super-cluster formation. a** ACE2-GFP cells (top panels) or ACE2-GFP cells incubated with spike/mCherry cells for 5 min (bottom panels) were photobleached (the box-indicated areas, **b**), and imaged at the indicated time points (sec.). Scale bars, 5 μm. **b** Quantification of the experiments in (**a**). Error bars indicated means ± SD. *n* = 8 cells. **c** ACE2-GFP cells incubated with spike/mCherry cells were stained with anti-spike antibodies (magenta). Shown are examples of 2D and 3D views, representing three biological repeats. Scale bars, 5 μm. **d** ACE2-GFP cells were incubated with spike/mCherry cells in the absence or presence of purified spike RBD for 10 min before imaging. Arrowheads indicate clustered ACE2. Scale bars, 5 μm. The graph shows the mean percentage of ACE2 cluster-positive cells from two biological repeats. **e** WT or SCL35B2 KO ACE2-GFP cells were incubated with spike/mCherry cells for 10 min before fixing and imaging. mCherry positive cells are shown in a surface-rendered view (top panels) and by the dashed lines (bottom panels). The arrowheads indicate ACE2 clusters at the cell–cell contact sites. **f** as in (**e**) except that WT ACE2-GFP cells were incubated with spike/mCherry cells in the absence or presence of PIXN (1 μM). ACE2-GFP clusters are shown in surface-rendered view in blue. The position of the spike/mCherry cells are indicated by the dashed lines. Scale bars, 5 μm. **g, h** Quantification of the size distribution of the ACE2 clusters in (**e**) and (**f**), respectively. **, *p* = 0.009; ***, *p* = 0.0009 by two-sided unpaired student's *t* test. *n* = 3 biological repeats. **i** ACE2-GFP cells incubated with spike/mCherry cells in the absence or presence of 1 μM PIXN were subject to 4D confocal imaging. The volume (Vol.) of ACE2 clusters at randomly selected cell–cell contact sites were analyzed by Imaris. Source data are provided as a Source Data file.

**LC1539.** $^1$H NMR (600 MHz, DMSO-$d_6$) δ 13.49 (s, 2H), 10.45 (t, *J* = 6.5 Hz, 2H), 7.67 (s, 2H), 7.22 (s, 2H), 3.82 (q, *J* = 6.5 Hz, 4H), 3.06 (d, *J* = 8.8 Hz, 4H). HRMS (ESI⁺): m/z calcd for $C_{18}H_{21}N_4O_4$ [M + H]⁺ 357.1557, found 357.1559.

**LC1540.** $^1$H NMR (600 MHz, CD₃OD) δ 9.04 (d, *J* = 1.8 Hz, 1H), 8.67 (d, *J* = 5.0 Hz, 1H), 7.74 (d, *J* = 4.9 Hz, 1H), 6.99 (dd, *J* = 4.2, 2.3 Hz, 2H), 3.75 (q, *J* = 3.5 Hz, 4H), 3.45 (dt, *J* = 13.0, 6.5 Hz, 4H), 2.95 (td, *J* = 6.5, 3.6 Hz, 4H), 2.87 (td, *J* = 5.5, 1.6 Hz, 4H). HRMS (ESI⁺): m/z calcd for $C_{21}H_{28}N_5O_4$ [M + H]⁺ 414.2136, found 414.2132.

**LC1542.** $^1$H NMR (600 MHz, CD₃OD) δ 7.38 (s, 2H), 7.32 (s, 2H), 3.92 (s, 6H), 3.70 (t, *J* = 5.5 Hz, 4H), 3.55 (t, *J* = 6.4 Hz, 4H), 2.98 (t, *J* = 6.5 Hz, 4H), 2.83 (t, *J* = 5.5 Hz, 4H). HRMS (ESI⁺): m/z calcd for $C_{24}H_{33}N_4O_6$ [M + H]⁺ 473.2395, found 473.2395.

**LC1553.** $^1$H NMR (600 MHz, CD₃OD) δ 7.27 (d, *J* = 9.2 Hz, 1H), 7.17 (d, *J* = 9.7 Hz, 1H), 7.07 (d, *J* = 9.8 Hz, 1H), 7.02 (d, *J* = 9.1 Hz, 1H), 3.87 (s, 3H), 3.71 (dt, *J* = 7.7, 5.5 Hz, 4H), 3.49 (t, *J* = 6.5 Hz, 2H), 3.44 (t, *J* = 6.5 Hz, 2H), 2.97 (t, *J* = 6.5 Hz, 2H), 2.93 (t, *J* = 6.5 Hz, 2H), 2.83 (dt, *J* = 11.6, 5.6 Hz, 4H). HRMS (ESI⁺): m/z calcd for $C_{23}H_{31}N_4O_6$ [M + H]⁺ 459.2238, found 459.2233.

**LC1519.** $^1$H NMR (600 MHz, CD₃OD) δ 6.81 (s, 2H), 6.75 (s, 2H), 3.70 (t, *J* = 4.5 Hz, 4H), 3.65 (t, *J* = 5.1 Hz, 4H), 3.58 (t, *J* = 4.5 Hz, 4H), 3.34–3.26 (m, 4H), 2.91–2.83 (m, 8H). HRMS (ESI⁺): m/z calcd for $C_{26}H_{37}N_4O_8$ [M + H]⁺ 533.2606, found 533.2601.

**LC1520.** $^1$H NMR (600 MHz, CD₃OD) δ 6.91 (s, 2H), 6.90 (s, 2H), 3.69–3.61 (m, 16H), 3.54 (t, *J* = 4.8 Hz, 4H), 3.37 (t, *J* = 6.6 Hz, 4H), 2.92–2.84 (m, 8H). HRMS (ESI⁺): m/z calcd for $C_{30}H_{45}N_4O_{10}$ [M + H]⁺ 621.3130, found 621.3136.

**LC1521.** $^1$H NMR (600 MHz, CD₃OD) δ 7.02 (s, 2H), 6.95 (s, 2H), 3.68–3.62 (m, 16H), 3.62–3.57 (m, 8H), 3.52 (t, *J* = 4.5 Hz, 4H), 3.44 (t, *J* = 6.9 Hz, 4H), 2.92 (t, *J* = 6.6 Hz, 4H), 2.88 (t, *J* = 5.1 Hz, 4H). HRMS (ESI⁺): m/z calcd for $C_{34}H_{53}N_4O_{12}$ [M + H]⁺ 709.3654, found 709.3648.

## Chemoenzymatic synthesis of HS 6-mers

A total of seven 6-mers were synthesized in this study using the chemoenzymatic synthetic approach[52]. These 6-mers are differed in the number of sulfo groups as well the presence or absences of 2-*O*-sulfated iduronic acid (IdoA2S) residue. All synthesis was initiated from glucuronide para-nitrophenyl, which is commercially available (Carbosyn). To synthesize the 6-mers without an IdoA2S residue, the synthesis involved the use of heparosan synthase 2 from Pasteurella multocida (pmHS2) and UDP-GlcNAc (or UDP-GlcNTFA, NTFA represents *N*-trifluoroacetylated glucosamine was incubated with pmHS2 (30 mg) and UDP-GlcNAc (3 mM) in a 100 mL of the reaction buffer containing 25 mM Tris-HCl (pH 7.2), 5 mM MnCl₂. The reaction was incubated at 37 °C overnight, and the 2-mer product was purified by a C-18 reverse phase column. The 2-mer product was further elongated to 3-mer in the 100 mL reaction buffer (pH 7.2) containing UDP-GlcA and purified on a C-18 column. The elongation and purification steps were repeated until the desired 6-mer was achieved. The final compound was confirmed for structural identity with Mass Spec and purity was checked with analytical HPLC. The pmHS2 enzyme, UDP-GlcA and UDP-GlcNTFA were made according to the protocol described in a prior publication[53,54]. Additional modification steps, including *N*-sulfation, 6-*O*-sulfation were completed using N-sulfotransferase and 6-*O*-sulfotransferase isoform 3, respectively[54]. To install an IdoA2S residue, 2-*O*-sulfotransferase and C5-pimerase were employed. The 3-*O*-sulfation to prepare NS2S6S3S 6-mer, 3-*O*-sulfotransferanse was used. The purity of the products was confirmed by high resolution anion exchange HPLC, and the molecular weight (MW) was determined by electrospray ionization mass spectrometry. As shown below, the purity of the 6-mers was in the range of 92% to 99%, and the measured MW was very close to the calculated MW (Calc MW).

| ID | Abbreviated sequence | Calc MW | Measured MW | Purity by HPLC |
|---|---|---|---|---|
| 0S | GlcNAc-GlcA-GlcNAc-GlcA-GlcNAc-GlcA-pNP[a] | 1277 | 1277 | 99% |
| NS | GlcNS-GlcA-GlcNS-GlcA-GlcNS-GlcA-pNP | 1391 | 1391 | 99% |
| 2S | GlcNS-GlcA-GlcNS-IdoA2S-GlcNS-GlcA-pNP | 1471 | 1470 | 97% |
| NS6S | GlcNS6S-GlcA-GlcNS6S-GlcA-GlcNS6S-GlcA-pNP | 1631 | 1630 | 94% |
| NS2S6S3S | GlcNS6S-GlcA-GlcNS6S3S-IdoA2S-GlcNS6S-GlcA-pNP | 1791 | 1791 | 92% |
| NS6S(6) | GlcNAc6S-GlcA-GlcNAc6S-GlcA-GlcNS6S-GlcA-pNP | 1555 | 1554 | 91% |
| NS6S(7) | GlcNS6S-GlcA-GlcNAc6S-GlcA-GlcNAc6S-GlcA-pNP | 1555 | 1554 | 99% |

[a]pNP refers to *para*-nitrophenyl group.

## Chemicals, reagents, cells, and viruses

A list of chemicals, proteins, antibodies, and other reagents used in the study is provided in Supplementary Table 2. 293 T cells stably expressing GFP-tagged human ACE2 (ACE2-GFP) were reported previously[11]. To make *SLC35B2* KO 293 T cells expressing ACE2-GFP, *SLC35B2* CRISPR KO cells[33] were transfecting cells with pCMV-ACE2-GFP (Codex Biosolution). Likewise, a plasmid expressing ACE-GS-GFP mutant was transfected into 293 T cells to make a stable line expressing GFP-tagged ACE2-GS mutant (Codex). GFP-positive cells were sorted by FACS after neomycin (1 mg/mL) selection for 1 week. These cells were maintained in Dulbecco's modified Eagle's medium (DMEM) supplemented with 10% fetal bovine serum (FBS), 1% penicillin/streptomycin. The in-house-generated stable lines will be available upon completion of a Material Transfer Agreement.

Calu-3 cells (HTB-55) and Vero E6 cells (CRL-1586) were purchased from ATCC. Vero TMPRSS2-E6 (TE6) (78081), a Vero E6-based cell line

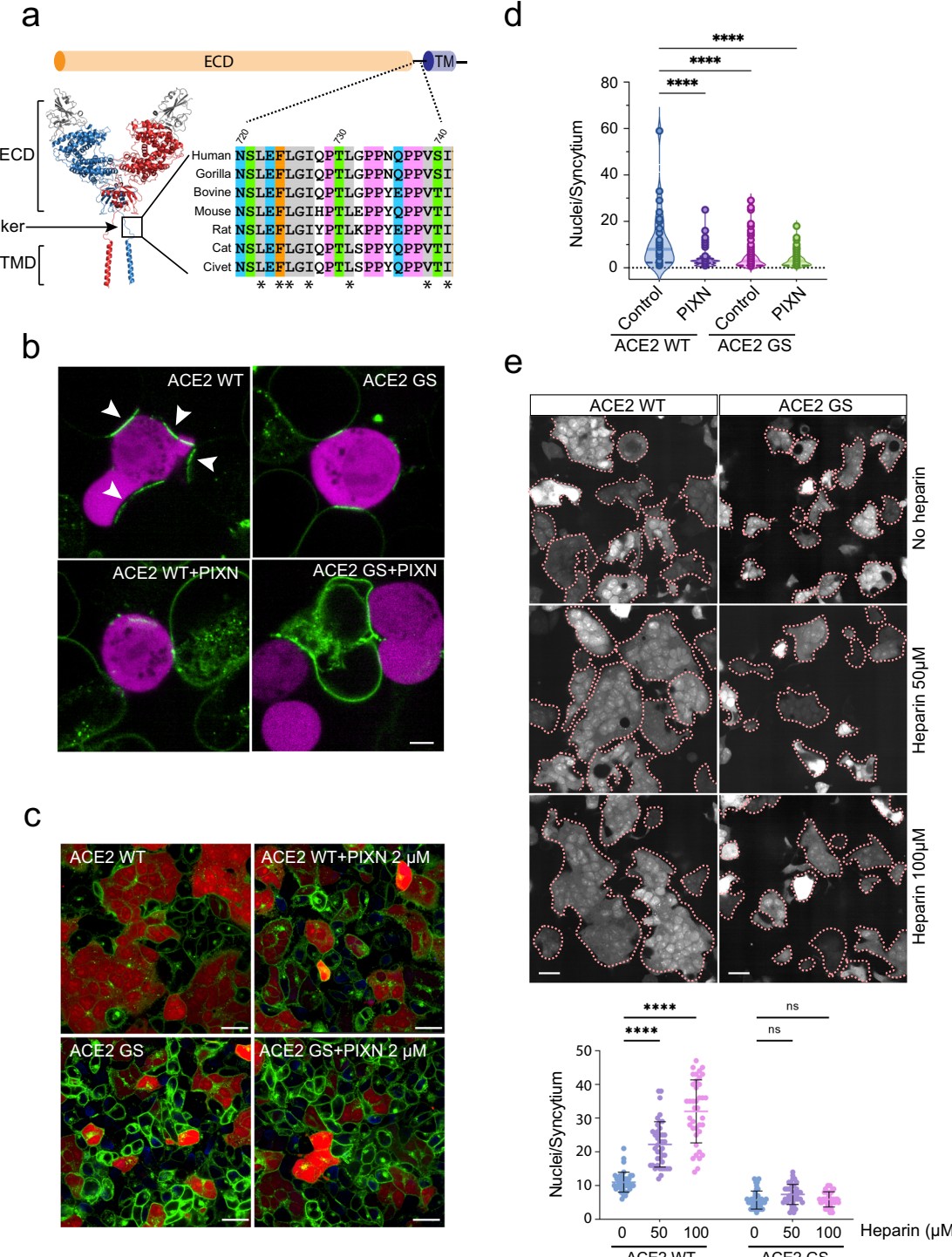

**Fig. 6 | A conserved ACE2 linker mediates spike-induced receptor clustering and syncytium formation. a** A schematic diagram of the ACE2 structure showing a conserved extracellular linker. ECD, extracellular domain; TMD, transmembrane domain. The asterisk indicates conserved hydrophobic residues. **b** ACE2-GFP or ACE2-GS-GFP cells were incubated with spike/mCherry cells (magenta) in the absence (top panels) or presence of PIXN (1 μM) (bottom panels) for 10 min before imaging. Note that ACE2 clustering enriches ACE2 signals at cell-cell contact sites (arrowheads), depleting it from other cell surface areas in ACE2 WT-GFP cells. Scale bar, 5 μm. Images are representative of two biological repeats. **c, d** The ACE2 linker promotes spike-induced cell−cell fusion. Cells stably expressing WT ACE2-GFP or the ACE2-GS-GFP mutant were incubated with spike/mCherry cells in the absence

(left panels) or presence of 2 μM PIXN for 60 min before imaging. Representative images are shown in (**c**). Scale bars, 30 μm. Quantification of the experiments is shown in (**d**). ****, $p < 0.0001$ by one-way ANOVA. $n = 3$ biological repeats. **e** Heparin-stimulated cell−cell fusion requires the ACE2 linker. WT ACE2-GFP cells or ACE2-GS-GFP cells were incubated with spike/mCherry cells in the presence heparin at the indicated concentrations for 60 min. Shown are representative images of the mCherry channel. Dashed lines indicate the boundary of the syncytia. Scale bars, 30 μm. The graph shows the quantification of the experiments. ****, $p < 0.0001$ by one-way ANOVA. $n = 3$ biological repeats. Source data are provided as a Source Data file.

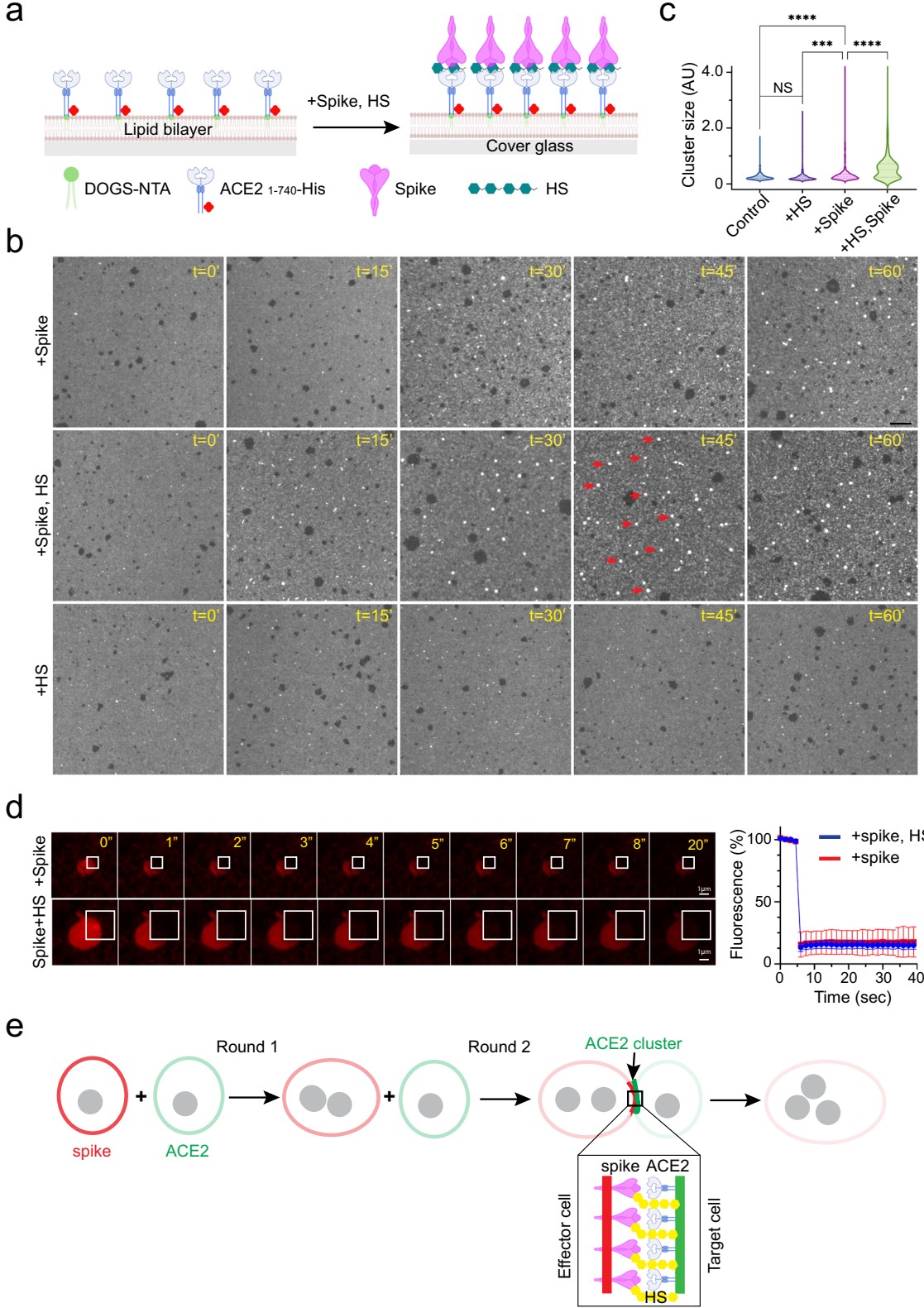

**Fig. 7 | HS facilitates spike-induced ACE2 clustering in a cell free system. a** A schematic illustration of the in vitro ACE2-clustering assay. Created by BioRender.com. **b**, **c** HS stimulates ACE2 cluster formation in the presence of the spike. Fluorescence-labeled ACE2 1-740 anchored to lipid bilayers in an imaging chamber was incubated with spike (15 nM) and HS (2.5 μM) either individually or in combination at 37 °C for the indicated time points. Arrows indicate examples of ACE2 clusters. Scale bar, 10 μm. **c** Quantification of the ACE2 cluster size in (**b**). ****, $p < 0.0001$ by one-way ANOVA. $n = 3$ biological repeats. **d** ACE2 is immobile in in vitro formed clusters. The indicated area in an ACE2 cluster formed in the

presence of spike (top panels) or spike combined with HS (bottom panels) were photobleached and imaged. The graph shows the GFP intensity in the indicated area over time. Error bars indicate means ± SD. $n = 6$ (with spike and HS) and 4 (with spike only) clusters, respectively. Scale bars, 1 μm. **e** HS-assisted ACE2 clustering helps overcome fusion-induced dilution of the spike, maintaining high levels of the fusogen at the fusion site. For simplicity, the box suggests a hypothetical model for HS-assisted ACE2 clustering, but other models are possible. Created by BioRender.com. Source data are provided as a Source Data file.

stably transfected with human TMPRSS2 was obtained from BPS Bioscience (San Diego, CA) and was maintained in Gibco high-glucose DMEM supplemented with 10% FBS, 1% penicillin/streptomycin, 10 mM HEPES pH 7.3, 1% Na pyruvate plus 3 μg/mL of Puromycin in a 37 °C incubator supplemented with 5% $CO_2$. Vero E6-TMPRSS2-T2A-ACE2 (TA6) (NR-54970), a Vero E6 cell line overexpressing both human TMPRSS2 and human ACE2 was obtained from BEI Resources (Manassas, VA) and was maintained in Gibco high-glucose Dulbecco's modified Eagle's medium (DMEM) supplemented with 10% FBS, 1% penicillin/streptomycin, 10 mM HEPES pH 7.3 plus 10 μg/mL of Puromycin in a 37 °C incubator supplemented with 5% $CO_2$. Cell lines were authenticated at their source prior to acquisition. No further authentication was performed.

The seeds of SARS-CoV-2 clinical isolates−USA-WA1/2020 and the Delta variant were obtained through BEI Resources (Manassas, VA). All seed viruses were amplified in TE6 cells in the infection medium (DMEM supplemented with 3% FBS) at 37 °C, 5% $CO_2$ for 3 days. Amplified viruses were aliquoted and stored in a secured −80 °C freezer until use. Virus titers were titrated in TE6 cells using an ELISA-based 50% tissue culture infectious dose ($TCID_{50}$) method[55,56].

### Infection with live SARS-CoV-2 in cultured cells

Vero TA6 cells were pre-seeded into eight-chamber slides (ibidi #80800) overnight. The next day after the aspiration of the growth media, cells were pre-treated with PIXN, MTAN, or DMSO diluted in the infection medium at 37 °C for 30 min. Then, without removing the compounds, cells were infected with USA-WA1/2020 or Delta variant (MOI of 0.1 or 0.5) at 37 °C, 5% $CO_2$ for another four 4 h before fixation and staining. For heparinase I/III treatment, cells were treated with a Heparinase I and III mixture that contains 180 μL DMEM medium, 20 μL 10x digestion buffer (NEB), 1.6 μL heparinase I, 4.0 μL heparinase II at 37 °C for 3 h. Treated cells were washed once with the medium and then subject to infection or spike-induced cell-cell fusion. Cells were fixed with 4% paraformaldehyde in Phosphate Buffer Saline (PBS) for 10 min followed by four washes with PBS before antibody staining. To confirm the removal of HS, we incubated cells with recombinant CFP+ at -100 nM on ice for 10 min before fixation and imaging.

### Infection with live SARS-CoV-2 in mice

B6.Cg-Tg(K18-ACE2)2Prlmn/J (K18-hACE2) transgenic mice (JAX Stock No. 034860)[36] were bred at 65–75 °F with 40–60% humidity at FDA White Oak Vivarium with 12-h light and dark cycles. The genotypes of the mice were individually confirmed (Transnetyx) before experiments. Age-matched male and female K18-hACE2 adult mice (the male and female ratio was approximately 1:1) at 12-16 weeks of age were injected intraperitoneally with 0.1 mL/mouse of PIXN at a final dose of 50 or 100 μg/kg, respectively. PIXN was dissolved in sterile PBS (pH 7.4) containing 1% of Penicillin/Streptomycin. K18-hACE2 mice receiving 0.1 mL/mouse of sterile PBS (pH7.4) containing 1% of Penicillin/Streptomycin served as controls. Two hours after injection, mice were inoculated intranasally with 1000 $TCID_{50}$/50 μL/mouse of live infectious USA-WA1/2020. Three days after infection, mice were euthanized, and whole lungs were harvested for viral load determination. Lungs were homogenized, and total RNA was extracted using the RNeasy Plus Mini Kit. The copies of the viral nucleocapsid (N) gene in homogenized lung tissues were amplified using the High-Capacity cDNA Reverse Transcription Kit and QuantiNova SYBR Green PCR kit in combination with 500 nM of 2019-nCoV RUO Kit according to the following program: 95 °C for 120 s, 95 °C for 5 s and 60 °C for 18 s (50 cycles) 1,3. A value of one was assigned if gene copies were below the detection limits. The mouse infection experiments with live SARS-CoV-2 were performed in an FDA Animal Biosafety Level-3 (ABSL-3) laboratory equipped with advanced access control devices and by trained personnel equipped with powered air-purifying respirators. All animal experiments were performed according to the procedures approved by the FDA White Oak Animal Program Animal Care and Use Committee.

### Pseudoviral particle entry assay

HEK293T-ACE2-GFP cells were seeded in white, transparent bottom 96-well microplates at 20,000 cells per well in 100 μL growth medium and incubated at 37 °C with 5% $CO_2$ overnight (-16 h). The growth medium was carefully removed, and 50 μL PP or PP-containing compounds were added to each well. The plates were then spinoculated by centrifugation at $453 \times g$ for 45 min and incubated for 24 h (48 h for Calu-3 cells) at 37 °C, 5% $CO_2$ to allow cell entry of PP and the expression of luciferase. After incubation, the supernatant was carefully removed. Then 50 μL/well of Bright-Glo luciferase detection reagent (Promega) was added to assay plates and incubated for 5 min at room temperature. The luminescence signal was measured by a Victor 1420 plate reader (PerkinElmer). For ACE2-GFP cells, the GFP signal was also determined by the plate reader. Data were normalized with wells containing PP but no compound as 100%, wells mock-treated with phosphate buffer saline (PBS) as 0%, and the ratio of luciferase to the corresponding GFP intensity was calculated.

### SARS-CoV-2 infection in a 3D EpiAirway model

Human bronchial epithelial cells (HBEC's 3D-EpiAirway™) were seeded into culture inserts for six-well plates one day before viral infection. Before adding drugs or virus, accumulated mucus from the tissue surface was removed by gently rinsing the apical surface twice with 400 μL TEER buffer. All fluids from the tissue surface were carefully removed to leave the apical surface exposed to the air. MTAN was diluted into the assay medium and placed at room temperature before co-treatment with a virus (MOI of 0.1) onto the apical and basal layers for one h. Following one h treatment, the virus was removed from the apical layer. The basolateral medium was replaced with fresh maintenance medium and compound at 24 h, 48 h, and 72 h post-infection.

At 24 h and 96 h post-infection, the apical layer was washed with 0.4 mL of the TEER buffer (PBS with Mg2+ and Ca2+). The washes were combined and aliquoted into separate microfuge tubes (1.5 mL). Eightfold serial dilutions of apical layer supernatant sample concentrations were added to 96-well assay plates containing Vero E6 cells (20,000/well). The plates were incubated at 37 °C, 5% $CO_2$, and 95% relative humidity. Following three days (72 ± 4 h) incubation, the plates were stained with crystal violet to measure the cytopathic effect. Virus titers were calculated using the method of Reed and Muench. The $TCID_{50}$ values were determined from duplicate samples.

### LDH assay

Medium from the basolateral layer of the 3-D tissue culture inserts was removed 24- and 96-h post-infection and diluted in an LDH Storage Buffer per the manufacturer's instructions (LDH-Glo Cytotoxicity Assay, Promega). Samples (5 μL) were further diluted with the LDH Buffer (95 μL) and incubated with an equal volume of LDH Detection Reagent. Luminescence was recorded after 60 min incubation at room temperature. As a negative control, we included a no-cell sample in determining the culture medium background. We used tissues treated with the apoptosis-inducing drug bleomycin as a positive control.

### ATP-based cytotoxicity assay

HEK293T-ACE2-GFP cells were seeded in a white, transparent bottom 96-well microplate (Thermo Fisher Scientific) at 20,000 cells per well in 100 μL growth medium and incubated at 37 °C with 5% $CO_2$ overnight (-16 h). The growth medium was carefully removed, and 100 μL medium with compounds was added into each well. The plates were then incubated at 37 °C for 24 h (48 h for Calu-3 cells) at 37 °C 5% $CO_2$. After incubation, 50 μL/well of ATPLite (PerkinElmer) was added to assay plates and incubated for 15 min at room temperature. The luminescence signal was measured using a Victor plate reader

(PerkinElmer). Data were normalized with wells containing cells but no compound as 100% and wells containing media-only as 0%.

## NMR study

NMR spectra were recorded at 313 K with a Bruker Avance III spectrometer operating at 850 MHz and equipped with a high sensitivity 5 mm TCI cryoprobe. Samples were lyophilized twice to remove residual solvents and were then dissolved in D2O (99.996%, Sigma, Co.) and placed in 5 mm NMR tubes. For the experiments involving free ligands, the samples were prepared to obtain a final concentration of $1.2 \times 10^{-3}$ M by dissolving hexasaccharide (1 mg) and Pixantrone (0.2 mg) in D2O. Proton spectra were recorded using water presaturation with a recycle delay of 10 s and 24 scans.

HSQC (heteronuclear single quantum coherence) experiments were performed in phase-sensitive mode with Echo/Antiecho-TPPI gradient selection using decoupling during acquisition and multiplicity editing during the selection step.

Thirty-two dummy scans and 20 scans with decoupling during the acquisition period with 1.5 s relaxation delay were accumulated, using a matrix size of $2048 \times 256$ datapoints.

Bidimensional HSQC-TOCSY data were acquired by using 20 scans per increment using a $2048 \times 256$ datapoints matrix with zero-filling in F1 to $2048 \times 2048$ points. 1H-13C HSQC-TOCSY were performed using 1.5 s relaxation delay and 100 ms mixing time.

For NOE experiments, the samples were prepared by dissolving hexasaccharide (1 mg) and Pixantrone (0.1 mg) in D2O, reaching a molar ratio of hexasaccharide/Pixantrone 2:1. NOESY experiments were performed at 313 K. A total of 24 scans was collected for each free induction decay (matrix $2048 \times 256$ points), the data were zero-filled to $2048 \times 2048$ points before Fourier transformation, and mixing time values of 120 ms was used.

## Drug and protein binding studies

HS or HS 6-mers of different concentrations were added to MTAN or PIXN (50 μM). After a brief incubation, absorbance was measured from 500 nm to 700 nm by Nanodrop 2000 (Thermo Fisher Scientific). The change in OD650 was determined and fitted into a binding curve using GraphPrism 9.0.

To study the interaction of spike with ACE2 in vitro, we used spike (200 ng/well) in PBS to treat a high binding 96 well plate at 4 °C for 15 h. The spike-coated plate was washed with PBS once and then incubated with the TMS buffer containing 4% bovine serum albumin (BSA), 20 mM Tris-HCl, 7.4, 150 mM NaCl, 2 mM $MgCl_2$ 2 mM $CaCl_2$ for 3 h at 37 °C. The plate was washed once with a BSA-free TMS buffer and then incubated for 1 h with heparin (5 μM) together with PIXN (20 μM) in the TMLS buffer containing 20 mM Tris-HCl, 7.4, 50 mM NaCl, 2 mM $MgCl_2$ 2 mM $CaCl_2$, 1% BSA to allow heparin binding to the spike. The plate was washed with the TMLS buffer once and then incubated with ACE2 recombinant protein bearing an hFC tag at various concentrations at room temperature for 1 h in the TMLS buffer. The plate was washed three times with the TMS buffer and then incubated with HRP-conjugated protein A in the TMS buffer for 1 h. The plate was washed four times and then developed with TMB turbo substrate for 5–10 min before the addition of 1 M sulfuric acid to quench the reactions. The absorbance was measured at 450 nm.

## Spike- and ACE2-mediated cell fusion assay

293 T cells expressing spike protein and mCherry were generated by transfecting cells in a 3.5 mm culture dish with 2.0 μg of pcDNA3.1-SARS-CoV-2-Spike (From the Delta variant) and pLV-mCherry at 10:1 ratio for 24 h. ACE2-GFP cells were seeded in fibronectin-coated 8-well Ibidi glass-bottom chambers at 50,000/well on day 1. Cell−cell fusion was conducted 48 h later. For each fusion reaction, 50,000 spike/mCherry-transfected cells (effector cells) were added to the imaging chamber. Live cell 4D imaging was initiated ~1 min after adding the effector cells. The fusion reactions were stopped for immunostaining experiments by incubating cells with a fixing buffer containing 4% paraformaldehyde in PBS for 15 min at room temperature. Cells were then stained with a Hoechst 33,342 staining solution to label the nuclei or with primary and secondary antibodies diluted in a PBS-based staining buffer containing 10% FBS and 0.2% Saponin.

## Transmission electron microscopy

Spike/mCherry donor cells and ACE2-GFP acceptor cells were mixed at 1:1 ratio and incubated in suspension at 37 °C for 12 min to form spike- and ACE2-induced membrane synapses. Cells were then fixed in a mixture of 2.5% glutaraldehyde and 1% paraformaldehyde in 0.1 M phosphate buffer, pH 7.4, for 90 min at room temperature. Subsequently, samples were washed three times for 10 min each in 0.1 M sodium cacodylate buffer, pH 7.4, before being post-fixed in 1% $OsO_4$ and 1.5% $K_3Fe(CN)_6$ in 0.1 M cacodylate buffer for 60 min on ice. Next, samples were rinsed and washed two times for 10 min each in water and incubated with 1% uranyl acetate overnight at 4 °C. The following day samples were rinsed and washed in water for 10 min and gradually dehydrated through a graded ethanol series followed by propylene oxide. Samples were then infiltrated in a gradient mix of propylene oxide and resin (Embed 812 resin) before being infiltrated with three changes of pure resin and embedded in 100% resin and baked at 60 °C for 48 h. Ultrathin sections (65 nm) were cut on an ultramicrotome (Leica EM UT7), and digital micrographs were acquired on JOEL JEM 1200 EXII operating at 80 kV and equipped with an AMT XR-60 digital camera.

## In vitro ACE2 clustering assay

To prepare small unilateral vesicles (SUV), we used glass syringes to prepare a DOGS-NTA lipid mixture in a glass vial as follows: rinse a glass vial with chloroform, and then add ~1 mL of chloroform plus individual lipids (POPC, 2.98 mg; DOGS-NTA, 0.085 mg, PEG-5000 PE, 0.023 mg). The lipid mixture was dried with a stable flow of nitrogen and then in a vacuum desiccator for 2 h. Resuspend the dried lipids in 1.5 mL of PBS and vortex. Transfer the resuspension into two 1.5 mL conical microcentrifuge tubes. Freeze the lipid resuspension in liquid nitrogen and thaw immediately in a water bath at room temperature. Repeat the freeze-thaw for 30 cycles. The cloudy solution will become clear over the freeze-thaw cycles. The lipid resuspensions were centrifuged at $22,000 \times g$ for 45 min at 4 °C. The SUV-containing supernatant was collected in a clean tube.

Ibidi u-Slide eight well Glass Bottom chambers were soaked in 5% Hellmanex III for 24 h (pre-heated to 50 °C) overnight, rinsed extensively with ultrapure water, and air dried. The glass surface was then treated with NaOH 1 M for 1 h at 50 °C. Treated chambers were rinsed with ultrapure water and washed three times with the Basic buffer (HEPES pH 7.3 50 mM, NaCl 150 mM). Each well was treated with 200 μL Basic buffer containing 7 μL SUV at 37 °C for 1 h followed by three washes with the Basic buffer. After the last wash, we blocked the lipid-coated surface with 200 μL Clustering Buffer (Basic buffer plus BSA 1 mg/mL) at 37 °C for 20 min. Add His8-tagged, Alexa555-labeled ACE2 1-740 (~50 nM) in the Clustering Buffer to each well and incubate the chamber at 29 °C for 1 h to allow His-tagged ACE2 to attach to the lipid bilayer. Wash each well three times with the Signaling Buffer (HEPES pH 7.3 50 mM, NaCl 50 mM). At this point, determine the mobility of ACE2 by FRAP. Add the spike in the Clustering Buffer at a final concentration of 14 nM in the presence or absence of HS. ACE2 clustering was imaged at the indicated time point by a Nikon CSU-W1 SoRa microscope equipped with a temperature control enclosure.

## Immunostaining, imaging acquisition, processing, and statistical data analyses

To stain cells with antibodies, cells were fixed in PBS containing 4% paraformaldehyde for 20 min at room temperature. Cells were washed

three times with PBS and then permeabilized with PBS containing 0.2% saposin and 10% fetal bovine serum. Cells were then stained with primary antibody overnight at 4 °C followed by secondary antibodies (1:2000). The dilution factors for primary antibodies are: anti-HSPG, 1:50; anti-NP of SARS-CoV-2, 1:10,000; Rabbit anti-spike 1:1000; mouse anti-spike, 1:500; anti-DNAJC5, 1:200.

Fluorescence confocal images were acquired by a Nikon CSU-W1 SoRa microscope equipped with a temperature control enclosure and a $CO_2$ control. 3D or 4D image reconstructions and analyses were done by Imaris software (Licensed to NIH). Fluorescence intensity was analyzed by open-source Fiji software. To this end, images were converted to individual channels, and regions of interest were drawn for measurement. Statistical analyses were performed using either Excel or GraphPad Prism 8.0 and 9.0. $P$ values were calculated by Student's $t$ test using Excel or one-way ANOVA by GraphPad Prism 8.0 and 9.0. Linear curve fitting, nonlinear curve fitting, and $IC_{50}$ calculation were done with GraphPad Prism 8.0 and 9.0. For nonlinear fitting, the inhibitor vs. response–variable slope model or the exponential decay model was used. Images were prepared by Photoshop and Illustrator (Adobe). Data processing and reporting are adherent to the community standards.

## Data availability
The authors declare that all data supporting the findings of this study are available within the Article, Supplementary Information, or Source Data file. For quantifications, an Excel file with all measurements is included in the provided Source Data file, which can be downloaded from https://doi.org/10.6084/m9.figshare.23261402[57]. Original raw imaging data is available from the corresponding author upon reasonable request. Source data are provided with this paper.

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

## Acknowledgements

We thank L.C. Pedersen and R.M. Petrovich of the Structural Biology Core at the NIEHS/NIH for providing the spike and the RBD proteins, the NHLBI flow cytometry and the NIDDK Advanced Light Microscopy & Image Analysis Cores (ALMIAC) for services, M. Ho (NCI) for critical reading of the manuscript. This research was supported by the intramural research program of NIDDK (Y.Y.), of NCAT (W.Z.), and of FDA (H.X.), the Center for Drug Design, University of Minnesota (L.C.), NIH HL094463, HL144970, and NSF GlycoMIP (DMR-1933525) (J.L.).

## Author contributions

Q.Z. and Y.Y. conceived the study, designed and performed the experiments. Y.Y. and W.Z. supervised the study. W.T. and H. X. designed and performed the authentic viral infection and mouse studies. E.S., V.P., and J.L. synthesized HS 6-mers and performed the NMR analysis. E.J. and L.C. designed and synthesized MTAN derivatives. Z.S. performed the EM analysis. C.C. supervised the 3D EpiAirway study. L.S., P.G, M.X., I.P., B.L., W.H. assisted in data collection and analyses. Q.Z., H.X., and Y.Y. wrote the paper.

## Competing interests

J.L. is a founder and the chief scientific officer for Glycan Therapeutics. V.P. is a chief operating officer for Glycan Therapeutics. Both J.L. and V.P. have equities for the company. Other authors declare no competing interests.
