## [Peer Review File · Nature Communications]

REVIEWER COMMENTS

Reviewer #1 (Remarks to the Author):

Qi Zhang and colleagues investigated the role of heparan sulphate (HS) in SARS-CoV-2 spike mediated fusion and syncytium formation. They started by testing newly generated MTAN-related drugs to investigate their effect on SARS-CoV-2 infection and their toxicity. They selected PIXN (LC1541) as their best candidate. It binds specific sulphate groups on HS and inhibits SARS-CoV-2 entry and replication in different systems including primary air-liquid cells and K18-hACE2 mice. Next, they showed that HS depletion inhibits spike mediated cell-cell fusion using both drugs and knock-out cells. They investigated the mechanism underlying HS-mediated increased fusion, showing that HS does not promote Spike-ACE2 binding or S2' cleavage. Using cell-cell fusion and in vitro techniques they show that HS induce clustering of ACE2 and promotion of fusion pore formation.

Overall, the article is clear and provides interesting mechanistic data on the action of HS on ACE2 and SARS-CoV-2 fusion.

Main comments

1. Fig 1. H. For the mouse experiments it would have been interesting to include data on the clinical score for SARS-CoV-2 infections and comment on the toxicity of the drug in vivo.
2. Fig. 3 - Cell line infection experiments are indicated being done at 4h post infection at relatively high MOIs (MOI of 0.5 or 0.1 was calculated on Vero TMPRSS2 cells) using Vero cells overexpressing both human ACE2 and TMPRSS2. At 4h post-infection the authors observed large syncytia and clear N staining (Fig. 3). 4h post infection seems like a very early timepoint to detect extended cell-cell fusion. According to published data (for instance Koch et al. EMBO, 2021) there is no significant detection of N after 4h of infection in Vero cells and from our experience, syncytia formation generally starts only as early as 6-7h post infection in highly permissive cell lines. Is this observation due to the usage of VeroE6 expressing both human ACE2 and TMPRSS2 making them highly fusogenic? Could the cell-cell fusion observed be due to fusion from without?
3. The authors should comment on the choice of this 4h timepoint. The results should be confirmed at lower MOIs, possibly including a drug blocking viral replication to distinguish fusion from without and de-novo produced spike.
4. Fig. 6. The authors identified a region of ACE2 promoting HS mediated receptor clustering. It would have been interesting to test if this region of ACE2 also plays a role if viral entry in addition to its role on cell-cell fusion. As in figure 1 and 3 the authors used virus and pseudotypes to show that HS modulating drugs impact viral entry, it may be worth testing if the phenotype for cell-cell fusion with this ACE2 mutant also translates to differential viral entry.

Reviewer #2 (Remarks to the Author):

This submission provides interesting findings to support the hypothesis that heparin and HSPGs facilitate the coalescence of target-cell ACE2 with effector-cell spikes. This coalescence correlates with higher levels of spike-directed cell-cell fusion. The authors suggest that this higher cell-cell fusion correlates with more severe COVID, and thus hypothesize that drugs interfering with this HSPG activity could be useful in mitigating COVID. They test new HSPG-binding agents as SARS-CoV-2 antivirals and find that one, PIXN, is moderately antiviral at nontoxic concentrations.

The significance of the findings appears moderate but with additional research there may be potential for HSPG interference as an antiviral measure.

The findings are sophisticated and high quality. These include results in medicinal chemistry,

biochemistry and virology, and they include excellent imaging data. However, even with the abundant data, the mechanism by which heparin and HSPGs facilitate viral fusion seems unclear. The summary models of ACE2 clustering in Fig 7 are only partially supported by the data.

The major issues in this review are centered on authors' rationales, i.e., justifications for experiments, and on authors' interpretations of findings. The minor issues include suggestions for additional experiments and modifications of the text.

Major points:

1. The model is that HSPGs facilitate coalescence of ACE2 on target cells. However, when HSPGs are not present or are inhibited by bound PIXN, there are still some ACE2-spike synapses and also cell-cell fusions. This may arise because spikes themselves are clustered on surfaces of effector cells, independent of whether the spikes are opposed next to adjacent ACE2 on target cells. Spikes coalesce in lipid rafts in plasma membranes of infected cells and spike-transfected cells. The investigators should image the positions of spike proteins on plasma membranes both before and after co-cultivation of spike-expressing cells with target cells. Clustered spikes will likely cluster ACE2 upon contact of spike-expressing and ACE2-expressing cells.
2. The finding that ACE2 lacking the dimerization domain (ACE2-GS) fails to coalesce in response to HSPGs is interesting. However, the interpretation of the result should be broadened. One spike trimer can bind more than one ACE2, generating ACE2 coalescence. If ACE2 is dimeric, then higher-order spike trimer : ACE2 dimer zippering (coalescence) seems likely. Alternative models for ACE2 clustering should be put forward.
3. Premise – justification for the study appeals to the need for durably active spikes as syncytia expand, even in the face of spike “dilution” on the ever-increasing plasma membrane surface areas of syncytia. This is an interesting idea but it does not have any experimental support and it may not be correct to assume that spikes are not continuously synthesized as syncytia expand. There could be plenty enough continued spike synthesis to generate large syncytia. This appeal to spike dilution appears prominently in the intro and discussion sections and this reviewer feels it should be tempered or communicated in context of other equally or more credible views about syncytial developments.

Minor points:

1. Fig. 1H; very modest in vivo antiviral activity in K18-hACE2 mice; does PIXN reduce pathogenicity of infections in this model?
2. Fig. 3B; the assay is not specifically measuring “virus entry”; y-axis should be relabeled as “infection”
3. Fig 3CD; need to emphasize when PIXN was added (added after virus entry or before?) If before, the smaller syncytia could merely be a result of reduced infection at entry stage, not actually blockade of syncytia per se.
4. Fig 4, what specifically is “semi-fusion”? Is it different than hemi-fusion?
5. Fig 5E (and elsewhere); “HS is required...” One can agree that HS facilitates ACE2 clustering but is not “required”, consider rephrasing.
6. Fig. 6; ACE2-GS; should test whether ACE2-GS confers SARS-CoV-2 susceptibility to ACE2-negative cells, quantifying ACE2-GS receptor activity relative to ACE2-WT.
7. Fig. 7A and 7E; depictions may be communicating inaccurately; a single S trimer may bind more than one ACE2 dimer, resulting in ACE2 clustering. Consider revising the images to account for this possibility.
8. Line 161; there is no evidence that N punctae are virions, they could be aggregates of intracellular N protein.
9. Lines 166-170; syncytia come from free S proteins going to cell surfaces, and likely far less so from “fusion of viral envelope with plasma membrane” (line 167); this part of text is not interpreting findings in the conventional and well-known ways.
10. Lines 286-287; there is no rationale stated for the test to determine whether LS is critical for cell-cell fusion. Consider major point #2 as a possible rationale.
11. Fig 7; should incubate the immobilized ACE2 with both soluble RBD monomers and soluble spike trimers, to determine whether trimerized state of spike is needed for ACE2 clustering.

We thank the reviewers for their positive and constructive suggestions. We have done many additional experiments to address these comments, as explained below point-by-point.

Reviewer #1 (Remarks to the Author):

Qi Zhang and colleagues investigated the role of heparan sulphate (HS) in SARS-CoV-2 spike mediated fusion and syncytium formation. They started by testing newly generated MTAN-related drugs to investigate their effect on SARS-CoV-2 infection and their toxicity. They selected PIXN (LC1541) as their best candidate. It binds specific sulphate groups on HS and inhibits SARS-CoV-2 entry and replication in different systems including primary air-liquid cells and K18-hACE2 mice. Next, they showed that HS depletion inhibits spike mediated cell-cell fusion using both drugs and knock-out cells. They investigated the mechanism underlying HS-mediated increased fusion, showing that HS does not promote Spike-ACE2 binding or S2' cleavage. Using cell-cell fusion and in vitro techniques they show that HS induce clustering of ACE2 and promotion of fusion pore formation.

Overall, the article is clear and provides interesting mechanistic data on the action of HS on ACE2 and SARS-CoV-2 fusion.

Response: We thank this reviewer for his/her enthusiasm in our study.

Main comments

1. Fig 1. H. For the mouse experiments it would have been interesting to include data on the clinical score for SARS-CoV-2 infections and comment on the toxicity of the drug in vivo.

Response: When we examined the viral titer in the lungs of mice treated with PIXN or untreated after infection, we did weight the lungs. We noticed a small reduction in lung weight in mice treated with PIXN (see below), suggesting that the drug might have some side effect on the lung. Because of this, and also because pathology-based scoring is less quantitative and has a narrower dynamic range compared to viral titer measurement, we chose the latter to determine the effect of drug treatment on viral entry and replication in mice.

From the drug development perspective, the reviewer raised an excellent point. However, we would like to emphasize that the significance of our finding is the mechanism underlying spike-induced membrane fusion, which reveals how the virus co-opts a cell surface molecule to enhance spike's membrane fusion activity. We now add a sentence on page 6 to clarify this, which reads as "The low *in vivo* anti-SARS-CoV-2 activity may be due to PIXN binding to HS in non-targeting tissues, which would reduce its effective concentration in the lung, suggesting that further optimizations are required to advance HS inhibitors into clinics as an anti-viral agent."

The toxicity of Pixantrone (PIXN) has been extensively evaluated in animals and in clinics (PMID: 31997425; ref 32 in the manuscript). To confirm this, we treated mice with PIXN and

analyzed the serum aspartate aminotransferase (AST) level as an indicator of liver toxicity 3 days after drug injection. We did not notice any increase in the AST level in the serum (Supplementary Fig. 1c), consistent with the known safety profile of this drug.

2. Fig. 3 - Cell line infection experiments are indicated being done at 4h post infection at relatively high MOIs (MOI of 0.5 or 0.1 was calculated on Vero TMPRSS2 cells) using Vero cells overexpressing both human ACE2 and TMPRSS2. At 4h post-infection the authors observed large syncytia and clear N staining (Fig. 3). 4h post infection seems like a very early timepoint to detect extended cell-cell fusion. According to published data (for instance Koch et al. EMBO, 2021) there is no significant detection of N after 4h of infection in Vero cells and from our experience, syncytia formation generally starts only as early as 6-7h post infection in highly permissive cell lines. Is this observation due to the usage of VeroE6 expressing both human ACE2 and TMPRSS2 making them highly fusogenic? Could the cell-cell fusion observed be due to fusion from without?

Response: The reviewer is correct that we chose 4-hour post infection (hpi) because we could not detect significant de novo spike synthesis at this time point, yet the infection-induced cell-cell fusion could be observed (Supplementary Fig. 3d). When we stained cells with a spike antibody, we could detect some spike signal as clustered dots on the surface 4 hpi (Supplementary Fig. 3d, Video 1), whereas de novo synthesized spike at 6 hpi is mostly localized at a peri-nuclear ERGIC compartment (Supplementary Fig. 3d). This difference suggests that the spike positive signal on the cell surface 4 hpi is derived from viral membrane during viral entry.

As for the MOI chosen in this study, we used MOI of 0.1-0.5, which is in line with published studies. For example, the study by Koch et al. EMBO 2021 used a MOI of 0.2. The paper by Braga L et al. (Nature 2021, <https://doi.org/10.1038/s41586-021-03491-6>, Figure 3) used a MOI of 0.5. In the study by Blanco-Melo, D. and colleagues (Cell 2020), the authors used 0.2 as low MOI and 2 as high MOI.

One key difference between our study and previous studies on SARS-CoV-2-induced syncytia is in the viral strain. Apparently, most studies published before 2021 used early strains such as the Washington strain or a closely related one. In our hand, only the Delta strain generates giant syncytia 4 h post infection, whereas the Washington strain only generates medium size syncytia after a much longer incubation (9 h) (Supplementary Fig. 3b). We never observed any syncytia with more than 2 nuclei per cell without infection, suggesting that the cell-cell fusion cannot be due to fusion from without. A few cells with two nuclei in uninfected cells are apparently caused by cell division.

3. The authors should comment on the choice of this 4h time points. The results should be confirmed at lower MOIs, possibly including a drug blocking viral replication to distinguish fusion from without and de-novo produced spike.

Response: We now add a sentence on page 8 to explain the choice of this time point (no significant de novo spike synthesis at this early time point). We repeated the infection experiment at a low MOI (0.01) for 4 and 6 hours, respectively, and in both Vero E6 cells and the highly permissive Vero TA6 cells (expressing ACE2 and TMPRSS2). As expected, we saw very few syncytia. Nevertheless, there seems to be a trend that is consistent with Delta being more potent in inducing cell-cell fusion and TA6 being more permissive than Vero E6 cells in cell-cell fusion (Supplementary Fig. 3c).

4. Fig. 6. The authors identified a region of ACE2 promoting HS mediated receptor clustering. It would have been interesting to test if this region of ACE2 also plays a role if viral entry in addition to its role on cell-cell fusion. As in figure 1 and 3 the authors used virus and pseudotypes to show that HS modulating drugs impact viral entry, it may be worth testing if the phenotype for cell-cell fusion with this ACE2 mutant also translates to differential viral entry.

Response: We tested the entry of pseudovirus-coated with the spike of the Washington strain, which is mediated primarily by ACE2-mediated endocytosis. In ACE2-GS-GFP cells, the entry of this virus appears identical to that in WT ACE2-GFP cells (Supplementary Fig. 6e), suggesting that this loop is not required for ACE2-mediated endocytosis of SARS-CoV-2.

Reviewer #2 (Remarks to the Author):

This submission provides interesting findings to support the hypothesis that heparin and HSPGs facilitate the coalescence of target-cell ACE2 with effector-cell spikes. This coalescence correlates with higher levels of spike-directed cell-cell fusion. The authors suggest that this higher cell-cell fusion correlates with more severe COVID, and thus hypothesize that drugs interfering with this HSPG activity could be useful in mitigating COVID. They test new HSPG-binding agents as SARS-CoV-2 antivirals and find that one, PIXN, is moderately antiviral at nontoxic concentrations.

The significance of the findings appears moderate but with additional research there may be potential for HSPG interference as an antiviral measure.

The findings are sophisticated and high quality. These include results in medicinal chemistry, biochemistry and virology, and they include excellent imaging data. However, even with the abundant data, the mechanism by which heparin and HSPGs facilitate viral fusion seems unclear. The summary models of ACE2 clustering in Fig 7 are only partially supported by the data.

Response: We agree with this reviewer that from a drug development perspective, the potency of PIXN may not be significant enough to warrant clinical testing at this points. However, it is worth noting that the most important finding of our study is about the unexpected role of heparan sulfate in promoting spike-mediated cell-cell fusion. We show that 1) ACE2 forms a super-cluster upon spike engagement, which is dependent on heparan sulfate; 2) ACE2 super-clustering concentrates the spike at the membrane contact site, which

facilitates fusion pore formation; 3) Heparan sulfate acts through a conserved loop in ACE2 to promote receptor super-clustering. All these findings are novel, which significantly advance our understanding on the mechanism of spike-mediated membrane fusion.

The major issues in this review are centered on authors' rationales, i.e., justifications for experiments, and on authors' interpretations of findings. The minor issues include suggestions for additional experiments and modifications of the text.

Major points:

1. The model is that HSPGs facilitate coalescence of ACE2 on target cells. However, when HSPGs are not present or are inhibited by bound PIXN, there are still some ACE2-spike synapses and also cell-cell fusions. This may arise because spikes themselves are clustered on surfaces of effector cells, independent of whether the spikes are opposed next to adjacent ACE2 on target cells. Spikes coalesce in lipid rafts in plasma membranes of infected cells and spike-transfected cells. The investigators should image the positions of spike proteins on plasma membranes both before and after co-cultivation of spike-expressing cells with target cells. Clustered spikes will likely cluster ACE2 upon contact of spike-expressing and ACE2-expressing cells.

Response: We thank the reviewer for this excellent suggestion. We have done the suggested experiment (Supplementary Fig. 5a). We stained spike transfected cells with a spike specific antibody either without or after co-culturing with ACE2 WT- or ACE2 GS-GFP cells. As pointed out by the reviewer, spike does form small coalesces on the cell surface, but these small puncta are uniformly distributed throughout the surface. By contrast, when spike cells encounter ACE2-GFP cells, the spike forms giant super-clusters that are co-localized with ACE2-GFP. Interestingly, when spike cells contact ACE2 GS-GFP cells, the spike clustering is significantly reduced (Supplementary Fig. 5a). This experiment suggests that it is ACE2 clustering that drives spike super-cluster formation, but not the other way around.

2. The finding that ACE2 lacking the dimerization domain (ACE2-GS) fails to coalesce in response to HSPGs is interesting. However, the interpretation of the result should be broadened. One spike trimer can bind more than one ACE2, generating ACE2 coalescence. If ACE2 is dimeric, then higher-order spike trimer : ACE2 dimer zippering (coalescence) seems likely. Alternative models for ACE2 clustering should be put forward.

Response: We agree with the reviewer that our data does not suggest how exactly ACE2 super-clusters are formed. We have now added more discussions on this point including a brief mentioning of the previous study on the role of heparan sulfate in facilitating FGF2 receptor dimerization (page 17). In this case, a structural study suggests that heparan sulfate can bind two FGF2 receptors to facilitate its dimerization. The linker segment may be involved in heparan sulfate-mediated ACE2 oligomerization in a similar way, but we do point out that "other alternative models cannot be excluded."

The zippering model suggested by the reviewer is also possible. However, in our initial attempt, we tried to reconstitute ACE2 super-clustering in solution, but failed to detect any

clustered ACE2 despite the use of a trimeric spike. This argues against a simple zippering model. Only after we attached the ACE2 receptor to lipid bilayers, did we see ACE2 clustering in the presence of the spike and heparan sulfate.

3. Premise – justification for the study appeals to the need for durably active spikes as syncytia expand, even in the face of spike “dilution” on the ever-increasing plasma membrane surface areas of syncytia. This is an interesting idea but it does not have any experimental support and it may not be correct to assume that spikes are not continuously synthesized as syncytia expand. There could be plenty enough continued spike synthesis to generate large syncytia. This appeal to spike dilution appears prominently in the intro and discussion sections and this reviewer feels it should be tempered or communicated in context of other equally or more credible views about syncytial developments.

Response: The reviewer is correct that during prolonged infection, there will be de novo synthesis of the spike. According to our time-course studies, we could not detect any spike expression in cells 4 hpi, yet in cells infected with the Delta strain, massive cell-cell fusion was seen, which is dependent on the viral concentration. At around 6 hours, we started to detect spike expression, which is mostly localized to the endoplasmic reticulum (ER) and some post ER vesicles (the ERGIC compartment), consistent with the presence of an ER retention signal in the C terminus of the spike. Given that the virus is assembled in a post-Golgi compartment akin to endolysosomes (PMID: 33157038), it is not surprising that most de novo synthesized spikes are localized to this compartment. In fact, it has been suggested by several studies that the deletion of the C-terminal 19 amino acids is necessary to re-distribute the spike to the cell surface (PMID: 32738193), which is the version widely used in cell-cell fusion studies including our study. Thus, under viral infection conditions, particularly during early stages of infection, there should be very little spike signal on the cell surface. We have revised the introduction to better explain this point on page 4. We also rephrase the question as “How spike maintains high fusogenic activities at low surface concentrations remains unclear”.

Additionally, we added more discussions (page 17). We specifically elaborate on the two-stage fusion model based on reviewer’s suggestion: the early phase induced by retribution of the viral spike to the cell surface during viral entry, and a second phase supported by de novo spike synthesis. Our data suggest that even for cell-cell fusion induced by de novo synthesized spike, as shown in HEK293T cell-based fusion assays (Figure 3), heparan sulfate is still involved and can function as a fusion enhancer. Thus, in either stage, concentrating spike at the fusion site, as facilitated by heparan sulfate, is a critical factor in spike-induced membrane fusion.

Minor points:

1. Fig. 1H; very modest in vivo antiviral activity in K18-hACE2 mice; does PIXN reduce pathogenicity of infections in this model?

Response: We agree with the reviewer that the effect of PIXN on viral entry and replication in vivo is modest, highlighting the necessity to further understand the PD and PK properties of

the drug. Given the relatively weak in vivo antiviral activity, it will be extremely difficult to demonstrate any significant improvement in lung morphology as this type of study is only semi-quantitative (See response to point 1 reviewer 1).

2. Fig. 3B; the assay is not specifically measuring “virus entry”; y-axis should be relabeled as “infection”

Response: We have changed the label to “Relative NP level (%)”.

3. Fig 3CD; need to emphasize when PIXN was added (added after virus entry or before?) If before, the smaller syncytia could merely be a result of reduced infection at entry stage, not actually blockade of syncytia per se.

Response: The drug was added together with the virus. We agree that this experiment by itself does not prove that heparan sulfate regulates cell-cell fusion per se. We change our conclusion to “our results raise the possibility that HS might enhance the fusogenic activities of the spike on the cell surface, which PIXN and MTAN antagonize”.

4. Fig 4, what specifically is “semi-fusion”? Is it different than hemi-fusion?

Response: hemi-fusion in general refers to a fusion stage in which the lipids in the outer leaflet of the donor and acceptor membranes are mixed, but fusion pore has not formed yet. This stage can either proceed towards the fusion if a fusion pore is formed or relax back to the unfused state. We used semi-fusion to refer to a subsequent stage in which a fusion pore is in forming, although it is not visible by our microscope. It is only indicated by the diffusion of the mCherry signal from the spike cells to ACE2 cells. We now revise the test to clarify this point on page 11.

5. Fig 5E (and elsewhere); “HS is required...” One can agree that HS facilitates ACE2 clustering but is not “required”, consider rephrasing.

Response: We have rephrased the sentence as suggested.

6. Fig. 6; ACE2-GS; should test whether ACE2-GS confers SARS-CoV-2 susceptibility to ACE2-negative cells, quantifying ACE2-GS receptor activity relative to ACE2-WT.

Response: We did show in the Supplementary Fig. 6c that ACE2-GS binds spike with a similar affinity as WT ACE2. To address the reviewer’s comment, we tested whether ACE-GS cells allow the entry of spike-coated pseudovirus. As shown in Supplementary Fig. 6e, the endocytosis-mediated entry of pseudovirus-coated by the spike was unaffected in ACE2-GS cells, suggesting that the dimerization function of this loop is only involved in membrane fusion, but dispensable for receptor-mediated endocytosis.

7. Fig. 7A and 7E; depictions may be communicating inaccurately; a single S trimer may bind more than one ACE2 dimer, resulting in ACE2 clustering. Consider revising the images to account for this possibility.

Response: As we do not know how exactly ACE2 forms super-clusters, we explain in the figure legend that the drawing only reflects one hypothetical model for simplicity, which does not exclude other modes of ACE2 oligomerization.

8. Line 161; there is no evidence that N punctae are virions, they could be aggregates of intracellular N protein.

Response: We have revised the text to clarify this point on page 8.

9. Lines 166-170; syncytia come from free S proteins going to cell surfaces, and likely far less so from “fusion of viral envelope with plasma membrane” (line 167); this part of text is not interpreting findings in the conventional and well-known ways.

Response: As explained above, our experimental design allows detection of infection-induced syncytia without significant de novo spike synthesis. Our data also suggest that regardless of the source of the spike (de novo synthesized vs. virally transferred), HS plays a critical role, enhancing the fusion efficiency. The text has been extensively revised to clarify this point.

10. Lines 286-287; there is no rationale stated for the test to determine whether LS is critical for cell-cell fusion. Consider major point #2 as a possible rationale.

Response: The rationale to test this domain in cell-cell fusion was simply because the sequence is so conserved evolutionarily, suggesting that it may have an unknown function that could be potentially exploited by the virus.

11. Fig 7; should incubate the immobilized ACE2 with both soluble RBD monomers and soluble spike trimers, to determine whether trimerized state of spike is needed for ACE2 clustering.

Response: We performed the proposed experiment and found that the RBD monomers are not capable of inducing ACE2 clustering in the presence of HS (Supplementary Fig. 7a).

REVIEWER COMMENTS

Reviewer #1 (Remarks to the Author):

The authors have addressed my concerns

Reviewer #2 (Remarks to the Author):

This resubmission contains some valuable findings that support a role for heparin and heparin sulfate in facilitating SARS-CoV-2 spike-mediated cell-cell fusion. Many of the experiments are sophisticated and the data quality are high. There are interesting findings that cell-cell fusion takes place at large micron-diameter areas in which spikes and receptors are opposed and clustered. There is evidence that heparin and HS can facilitate this clustering. There are data showing that a portion of ACE2, once changed, will eliminate the facilitating effect of ACE2. This is interesting. However, there are still many additional concerns about this paper. Several of the concerns are minor but collectively they indicate that more work and more revision would be needed to improve the report. Similar to the issues during the first round of review, this re-review has questions about the authors' interpretations of findings and their support for the statements made in the text. The concerns are listed below in the order in which they appear in the text.

1. Line 29, the mechanism of syncytium formation is not "largely unclear". There are many excellent papers describing the mechanisms.
2. Lines 37-38, "the interaction of HS with spike allosterically enables a conserved ACE2 linker...." Readers will be unclear about the meaning of this sentence in the abstract.
3. Lines 39, fusion is targeted by the HS-binding drug, but not "effectively".
4. Line 49, S1 and S2 are not covalently linked.
5. Line 52, the fusion peptide does not "drive fusion..". It is the collapse of the S2 fusion intermediates into helical bundles that drives fusion.
6. Line 70, what is the evidence that viruses, after fusion with plasma membranes, leave behind a cluster of fusion-active spikes?
7. Line 72, what is the evidence that spikes are redistributed on plasma membranes during virus entry, causing spike dilution? These statements are put forward as though they are known, but where are the supporting data?
8. Line 75, even if spikes were deposited on plasma membranes after virion-host cell membrane fusion, they would not then "limit the cell surface spike level". Further clarify.
9. Line 78, it is not known how "low" the spike concentrations are... many unsupported assumptions are here in the introduction.
10. Line 109, Fig 1d data appear to show that PIXN reduces the binding of virions to cells. How can it be claimed that PIXN suppresses endocytic virus entry?
11. Lines 110-116, Fig 1f and 1g show that PIXN is only antiviral at the highest 20 micromolar dose, but at this dose, the PIXN is cytotoxic, so the antiviral effect is likely due to direct PIXN cytotoxicity. This finding does not "show that PIXN has a favorable anti-SARS-CoV-2 activity in vitro".
12. Lines 117-128, PIXN in vivo antiviral activity is so weak that it is difficult to see the value of the findings here.
13. Lines 164-165, it does not appear that investigators "have established PIXN and MTAN as HS-binding drugs that block endocytosis-mediated entry of SARS-CoV-2". Precisely where is the evidence for this statement?
14. Lines 173-174, delta entry by plasma membrane is assumed but not shown, the statement on these lines should be tempered.
15. Line 176, you mean de novo "spike" synthesis, not synthesis in general.
16. Line 185-188, why would multiple rounds of fusion be required to fuse up to 80 cells? Were inoculated virions rinsed away after a 4C binding period? If not, then virions in media could keep binding to cells, generating syncytia via fusion-from-without.
17. Line 186-187, is it known that the spike molecules are "transferred from the viral particles"? Might virions bind between cells, forming virion bridges between cells, which then offer an inter-cellular fusion to occur via virions?
18. Line 191-192, does PIXN and MTAN reduce the binding of virions to cells, in possibly doing so, reducing the level of virion-based "fusion-from-without"? This would be an interpretation of findings

that does not raise the possibility that HS enhances the fusogenic activation fo the spike on the cell surface.

19. Lines 261-262, how can it be claimed that the red arrow is pointing to a fusion event? In the figure legend it is stated as though this is a fact, but in the text, one reads that the red arrow "probably points to a fusion event.

20. Lines 359-360, do spike antibodies block syncytia, if yes, then how do syncytia allow viruses to escape antibody-mediated neutralization?

21. Lines 365-367, what is the evidence that membrane fusion activity of spike can sustain over multiple rounds? Were virions bound to cells at 4C, then cells rinsed extensively, then incubated to the 4 hpi time point without residual inoculum in the media, or were virions applied to cells, leaving residual inoculum to continue binding to cells, allowing for continued fusion from without to occur?

22. Lines 380-381, what alternative models are envisioned? Consider expanding here.

23. ACE2-GS, does the portion of ACE2 replaced by GS include the ADAM17/TACE and TMPRSS2 cleavage site? Is this relevant to ACE2-GS activity?

24. Can there be some discussion of the findings in this paper with those in this recent paper that has been getting some press? Eiring et al, Coronaviruses Use ACE2 Monomers as Entry-Receptors, *Angewandte Chemie International Edition*(2023). DOI: 10.1002/anie.202300821

We thank this reviewer for careful reviewing of our manuscript. We have made changes to the text to clarify all the points.

1. Line 29, the mechanism of syncytium formation is not “largely unclear”. There are many excellent papers describing the mechanisms.

Response: We have revised the text. It now reads as “To better elucidate the mechanism of syncytium formation associated severe COVID-19, we.....”

2. Lines 37-38, “the interaction of HS with spike allosterically enables a conserved ACE2 linker...” Readers will be unclear about the meaning of this sentence in the abstract.

Response: Since the spike binds ACE2 at a site distal from the linker segment, we assumed that the activation of this linker segment during ACE2 clustering must occur by an allosteric mechanism. We agree that it may not be easy to get this point without reviewing the data. **We removed the word “allosterically” from the abstract.**

3. Lines 39, fusion is targeted by the HS-binding drug, but not “effectively”.

Response: We removed “effectively”.

4. Line 49, S1 and S2 are not covalently linked.

Response: We have corrected the text on page 3.

5. Line 52, the fusion peptide does not “drive fusion..”. It is the collapse of the S2 fusion intermediates into helical bundles that drives fusion.

Response: We changed the text to “inducing TMPRSS2-mediated cleavage of the S2 fragment that ultimately leads to the collapse of an S2 fusion intermediate into a helical bundle to drive the fusion of viral membranes with the plasma membrane.”

6. Line 70, what is the evidence that viruses, after fusion with plasma membranes, leave behind a cluster of fusion-active spikes?

Response: If all spike molecules were cleaved by TMPRSS2 during viral entry, given the short time window of the viral entry, the cleavage should be extremely efficient. This is INCONSISTENT with the fact that the virus has constantly evolved in a direction that allows the spike to be cleaved more and more efficiently by TMPRSS2 (Rajah, 2021, EMBO), suggesting that the S2 cleavage by TMPRSS2 is far from optimal. We now add a sentence on page 3 to highlight this point. Additionally, a recent study showed that even when both spike and TMPRSS2 are co-expressed in the same cell, the cleavage only occurs to a small population of spike (PMID: 32703818). Likewise, in our cell-cell fusion assay, we measured the cleavage of S2 by a chase experiment (Figure S5d, e). Even after 2 hours of incubation (all cells have fused at least once), only a small fraction of the S2 fragment was converted to S2'. Note that this experiment was conducted with the spike from the Delta variant. Thus, even for spike with a strong fusion activity, it is still cleaved very inefficiently. These observations all suggest that membrane fusion can take place with only a small fraction of spike being consumed by cleavage.

We added a sentence on page 3 to highlight the evolution of the virus, which is associated with increased spike cleavage efficiency.

7. Line 72, what is the evidence that spikes are redistributed on plasma membranes during virus entry, causing spike dilution? These statements are put forward as though they are known, but where are the supporting data?

Response: When a membrane-encircled virus fuses with host cells, the viral membrane would become part of the plasma membrane. This is a well-accepted concept in the membrane fusion field (see the cited review by Harrison, S. 2008 Nat. Strul. Mol. Biol. PMID: 18596815). Since spike is a membrane protein, it should be free to distribute to the entire cell surface by diffusion. We now add a sentence on page 4 to clarify this point. It reads as “Strikingly, when the virus fuses with the plasma membrane, the viral membrane becomes part of the cell membrane, leading to a drastic expansion of the spike-containing membranes.....”

8. Line 75, even if spikes were deposited on plasma membranes after virion-host cell membrane fusion, they would not then “limit the cell surface spike level”. Further clarify.

Response: What we meant was that the expansion of the spike-containing membranes during viral entry and the retention of the newly synthesized spike to the cell surface by the ER retention signal **collectively** limit the cell surface spike level. We have re-worded the sentences to avoid the confusion.

9. Line 78, it is not known how “low” the spike concentrations are... many unsupported assumptions are here in the introduction.

Response: We removed the phrase “at low surface concentration”.

10. Line 109, Fig 1d data appear to show that PIXN reduces the binding of virions to cells. How can it be claimed that PIXN suppresses endocytic virus entry?

Response: The reviewer is correct that we did not detect significant accumulation of the viral particles on the lateral cell surface in these confocal images of PIXN-treated cells (Fig. 1d). However, since we detected no effect of these drugs on the interaction of spike with ACE2 (Fig. S4c) and also because the interaction of spike-expressing cells with ACE2 cells was not affected by PIXN (Fig. S4d, e), we believe that PIXN does not disrupt the spike (and therefore the virus) binding to ACE2 cells. The lack of cell surface spike staining in drug-treated cells was most likely caused by a detection sensitivity issue or by our initial focus on confocal sections around endolysosomes. As we later revealed in Fig. S3d by 3D reconstruction, the viral particles on the cell surface are usually clustered on apical surface of the cell, which would escape detection by deeper confocal sections in Fig. 1d.

To avoid potential overinterpretation of the data, we have tempered our conclusion. The sentence reads as “suggesting that the drug inhibits viral entry at a step upstream of endocytosis”.

11. Lines 110-116, Fig 1f and 1g show that PIXN is only antiviral at the highest 20 micromolar dose, but at this dose, the PIXN is cytotoxic, so the antiviral effect is likely due to direct PXIN

cytotoxicity. This finding does not "show that PIXN has a favorable anti-SARS-CoV-2 activity in vitro".

Response: Please note that even at the lowest dose chosen (200 nM), there was a significant inhibition of viral replication when compared to the vehicle controls (the dotted lines). We have changed the text to make this point clear.

12. Lines 117-128, PIXN in vivo antiviral activity is so weak that it is difficult to see the value of the findings here.

Response: We agree with the reviewer that the in vivo efficacy of PIXN is not very strong. However, we feel that it is important to include the data to make the study complete. The result may provide a base for future optimization studies. Considering the reviewer's criticism, we now move this data to supplementary Fig. 1d, e.

13. Lines 164-165, it does not appear that investigators "have established PIXN and MTAN as HS-binding drugs that block endocytosis-mediated entry of SARS-CoV-2". Precisely where is the evidence for this statement?

Response: The binding of PIXN and MTAN to HS was shown in Figure 2 and related supplementary figures. The inhibition of SARS-CoV-2 entry was shown in Figure 1d. We now refer to these figures in the text.

14. Lines 173-174, delta entry by plasma membrane is assumed but not shown, the statement on these lines should be tempered.

Response: We now add an additional reference that shows increased membrane fusion activity of Delta spike. This study (Rajah, 2021, EMBO), combined with the paper cited in the previous version (Zhang, 2022 Science), suggests that delta strain should primarily use the plasma membrane as the entry site provided sufficient TMPRSS2 expression. We also point out that in our immunostaining experiment (Fig. 3a), the viral protein NP is not localized to the endolysosome, consistent with the notion that it uses the plasma membrane as the entry site.

15. Line 176, you mean de novo "spike" synthesis, not synthesis in general.

Response: The reviewer is correct. We have added "Spike" to the sentence.

16. Line 185-188, why would multiple rounds of fusion be required to fuse up to 80 cells? Were inoculated virions rinsed away after a 4C binding period? If not, then virions in media could keep binding to cells, generating syncytia via fusion-from-without.

Response: We now include a brief discussion about "fusion-from-without", a relatively uncommon mechanism for virus-induced cell-cell fusion. Specifically, we cited a recent paper in iScience (Theuerkauf, 2021 iScience), which demonstrate that for virus-like particles containing spike, it needs very high concentrations to induce fusion-from-without (5,000 VLP/cell). This would not happen under our experimental condition given that relatively low viral titer used.

17. Line 186-187, is it known that the spike molecules are "transferred from the viral particles"?

Might virions bind between cells, forming virion bridges between cells, which then offer an inter-cellular fusion to occur via virions?

Response: Again, the scenario described here (also referred to as fusion-from-without) should not happen under our experimental conditions as explained for point 16.

18. Line 191-192, does PXIN and MTAN reduce the binding of virions to cells, in possibly doing so, reducing the level of virion-based "fusion-from-without"? This would be an interpretation of findings that does not raise the possibility that HS enhances the fusogenic activation of the spike on the cell surface.

Response: We show in Supplementary Fig. 4c, d, e that neither PXIN nor MTAN affects the binding of spike to ACE2 or ACE2-expressing cells. Thus, these drugs should not affect virus binding to the cells. Additionally, as pointed out above, "fusion-from-without" does not happen under our experimental conditions.

19. Lines 261-262, how can it be claimed that the red arrow is pointing to a fusion event? In the figure legend it is stated as though this is a fact, but in the text, one reads that the red arrow "probably points to a fusion event."

Response: We added probably to the figure legend.

20. Lines 359-360, do spike antibodies block syncytia, if yes, then how do syncytia allow viruses to escape antibody-mediated neutralization?

Response: The statement here is based on the following article, which shows that viral spreading via syncytia is more resistant to neutralizing antibodies (Zeng 2022, PNAS). We have revised text. It now reads as "it was proposed that this transmission route allows the virus to escape immune surveillance and antibody-mediated neutralization".

21. Lines 365-367, what is the evidence that membrane fusion activity of spike can sustain over multiple rounds? Were virions bound to cells at 4C, then cells rinsed extensively, then incubated to the 4 hpi time point without residual inoculum in the media, or were virions applied to cells, leaving residual inoculum to continue binding to cells, allowing for continued fusion from without to occur?

Response: We have removed this speculative statement from the discussion.

22. Lines 380-381, what alternative models are envisioned? Consider expanding here.

Response: We mentioned that "How HS facilitates ACE2 super-cluster formation remains to be determined" on page 18. At this point, any models would be speculative in nature. We only mentioned one possibility based on previous structural studies on the role of HS in regulating FGF receptor oligomerization. The analogy raises the possibility that a similar mechanism may be applied to ACE2 oligomerization.

23. ACE2-GS, does the portion of ACE2 replaced by GS include the ADAM17/TACE and TMPRSS2 cleavage site? Is this relevant to ACE2-GS activity?

Response: The answer is No. The linker segment is from 721-740. The TMPRSS2 and ADAM17/TACE-mediated cleavage occurs between 697-716 (PMID: 24227843).

24. Can there be some discussion of the findings in this paper with those in this recent paper that has been getting some press? Eiring et al, Coronaviruses Use ACE2 Monomers as Entry-Receptors, *Angewandte Chemie International Edition*(2023). DOI: 10.1002/anie.202300821

Response: We now mentioned the article that suggests the possibility of ACE2 being a monomer, which if true would certainly impact our model. However, this new study challenges a well-established dogma in the field with only limited evidence (entirely imaging-based). The controversy certainly will require more studies to resolve.

REVIEWERS' COMMENTS

Reviewer #2 (Remarks to the Author):

The authors adequately responded to second-round reviewer questions.